# Circulating blood eNAMPT drives the circadian rhythms in locomotor activity and energy expenditure

Jae Woo Park [1], Eun Roh [2], Gil Myoung Kang[3], So Young Gil [3], Hyun Kyong Kim[3], Chan Hee Lee[4], Won Hee Jang [1], Se Eun Park[1], Sang Yun Moon[1], Seong Jun Kim[1], So Yeon Jeong[1], Chae Beom Park [1], Hyo Sun Lim[1], Yu Rim Oh [1], Han Na Jung [3,5], Obin Kwon [6], Byung Soo Youn[7], Gi Hoon Son [8], Se Hee Min [3,5] & Min-Seon Kim [3,5] ✉

Nicotinamide adenine dinucleotide (NAD$^+$) is an essential cofactor of critical enzymes including protein deacetylase sirtuins/SIRTs and its levels in mammalian cells rely on the nicotinamide phosphoribosyltransferase (NAMPT)-mediated salvage pathway. Intracellular NAMPT (iNAMPT) is secreted and found in the blood as extracellular NAMPT (eNAMPT). In the liver, the iNAMPT−NAD$^+$ axis oscillates in a circadian manner and regulates the cellular clockwork. Here we show that the hypothalamic NAD$^+$ levels show a distinct circadian fluctuation with a nocturnal rise in lean mice. This rhythm is in phase with that of plasma eNAMPT levels but not with that of hypothalamic iNAMPT levels. Chemical and genetic blockade of eNAMPT profoundly inhibit the nighttime elevations in hypothalamic NAD$^+$ levels as well as those in locomotor activity (LMA) and energy expenditure (EE). Conversely, elevation of plasma eNAMPT by NAMPT administration increases hypothalamic NAD$^+$ levels and stimulates LMA and EE via the hypothalamic NAD$^+$−SIRT−FOXO1−melanocortin pathway. Notably, obese animals display a markedly blunted circadian oscillation in blood eNAMPT−hypothalamic NAD$^+$−FOXO1 axis as well as LMA and EE. Our findings indicate that the eNAMPT regulation of hypothalamic NAD$^+$ biosynthesis underlies circadian physiology and that this system can be significantly disrupted by obesity.

The recent obesity pandemic can be attributed to the imbalance between the evolutionarily adapted biological system and the rapidly changing environmental factors. A good example is the induction of obesity by disruptions of the circadian rhythm. Human epidemiological studies have demonstrated that chronic desynchrony of internal circadian time with the external light-dark environment correlates with an increased incidence of obesity and type 2 diabetes[1,2]. Moreover, knockout of the *Clock* gene in mice induces the phenotypes of metabolic dysregulations, including overeating, obesity, and glucose intolerance[3], demonstrating a

[1]Department of Biomedical Science, Asan Medical Institute of Convergence Science and Technology, University of Ulsan College of Medicine, Seoul 05505, Korea. [2]Division of Endocrinology and Metabolism, Department of Internal Medicine, Hallym University Sacred Heart Hospital, Anyang 14068, Korea. [3]Appetite Regulation Laboratory, Asan Institute for Life Science, Seoul 05505, Korea. [4]Program of Material Science for Medicine and Pharmaceutics, Hallym University, Chuncheon 24252, Korea. [5]Division of Endocrinology and Metabolism, Diabetes Center, Asan Medical Center, University of Ulsan College of Medicine, Seoul 05505, Korea. [6]Department of Biomedical Sciences, Seoul National University College of Medicine, Seoul 03080, Korea. [7]OsteoNeuroGen, Seoul 08507, Korea. [8]Department of Biomedical Science, Korea University College of Medicine, Seoul 02841, Korea. ✉e-mail: mskim@amc.seoul.kr

causal relationship between disrupted clockwork and obesity/metabolic disorders.

The circadian rhythmicity of various physiological functions is primarily driven by the pacemaker neurons resided in the hypothalamic suprachiasmatic nucleus (SCN) that uses light as a primary cue[4]. On the other hand, subordinate clocks are distributed in a number of cells across the body[5–7]. The SCN master clock synchronizes the subordinate clocks to a uniform internal time via yet not-fully understood mechanisms and coordinates the circadian control of behaviors (e.g., feeding-fasting, sleep-wakefulness), neuroendocrine functions, and autonomic nervous systems[5–7]. At the molecular level, the cellular clockwork is composed of transcription–translation-based autoregulatory feedback loops[8,9]. The heterodimeric complexes of the key clockwork transcription factors CLOCK and BMAL1 bind to the promoter of clock genes (Period 1–3 and Cryptochrome 1/2) to stimulate their production. As clock genes act as a suppressor of CLOCK/BMAL1, clock gene production leads to the repression of their own transcription[8,9]. Such positive and negative regulations generate the cyclic oscillation of clock gene expression over a 24-h period.

Nicotinamide adenine dinucleotide (NAD)$^+$ is a critical metabolite that regulates biochemical reactions catalyzed by the NAD$^+$-dependent enzymes such as sirtuins (SIRT1–7), poly(ADP-ribose) polymerase (PARP), and cyclic ADP-ribose cyclase/CD38[10,11]. In mammals, cellular NAD$^+$ levels mainly depend on its salvage pathway, in which NAD$^+$ is resynthesized from nicotinamide via nicotinamide mononucleotide (NMN)[10–12]. Nicotinamide phosphoribosyltransferase (NAMPT) catalyzes the rate-limiting first step of NAD$^+$ salvage pathway and thus the inhibition of NAMPT activity leads to a significant depletion in cellular NAD$^+$ levels[10,12]. Intracellular NAMPT (iNAMPT) is secreted to the bloodstream in the form of extracellular vesicles[13] and taken by cells in remote organs. Interestingly, this secreted form of NAMPT, so-called extracellular NAMPT (eNAMPT), is superior to iNAMPT in terms of NAD$^+$ biosynthetic activity[14], thus raising the possibility of interorgan regulation of NAD$^+$ biosynthesis via eNAMPT. Supporting this possibility, a previous study showed that plasma eNAMPT and hypothalamic NAD$^+$ levels were significantly decreased by depleting the *Nampt* gene in adipocytes, which are shown to secrete eNAMPT[15].

NAD$^+$-dependent deacetylase SIRT1 is a master metabolic regulator that links the cellular energy status to adaptive transcriptional responses[16]. In the peripheral metabolic organs, SIRT1 modulates the cellular metabolic processes by deacetylating important transcriptional regulators, such as forkhead box protein O1 (FOXO1), peroxisome proliferator-activated receptor gamma (PPARγ), and PPARγ coactivator 1-alpha (PGC1α)[17–19]. SIRT1 is also essential for normal functions of hypothalamic neurons that are responsible for the homeostatic control of energy metabolism[11,20–23]. As the SIRT enzymatic activity depends on the cellular NAD$^+$ level, adequate NAD$^+$ levels in hypothalamic neurons are pivotal for maintaining energy homeostasis.

Aside from its role in metabolic regulation, the iNAMPT-NAD$^+$-SIRT1 axis is an indispensable regulator of cellular clockwork[24–27]. In the liver, cellular levels of iNAMPT and NAD$^+$ show dramatic circadian oscillations[27]. Moreover, SIRT1 binds the CLOCK:BMAL1 complexes in a circadian manner[26] and regulates the clock gene transcription by deacetylating BMAL1 and PER2[26–29]. Considering the importance of the NAMPT-NAD$^+$-SIRT1 axis in the peripheral tissue clockwork[27], we decided to study an involvement of the hypothalamic NAMPT-NAD$^+$-SIRT axis in the circadian regulation of energy metabolism. Here, we show that like in the liver, the NAD$^+$ levels in the mediobasal hypothalamus display a prominent circadian fluctuation, which is thought to be driven by blood eNAMPT. Moreover, circadian oscillations of plasma eNAMPT and hypothalamic NAD$^+$ levels coordinate the day-night rhythm in locomotor activity (LMA) and energy expenditure (EE).

## Results

### Nocturnal rhythmic elevation of hypothalamic NAD$^+$ levels is in phase with that of plasma eNAMPT

We first tested if the hypothalamic NAD$^+$ levels display circadian oscillations. For this purpose, we collected the mediobasal hypothalamus (MBH), which includes the SCN, hypothalamic arcuate nucleus (ARC), and ventromedial nucleus (VMN), from lean C57BL6 mice every 4 h throughout two consecutive 12 h-light/dark cycles and measured the NAD$^+$ levels in the MBH. We found a marked circadian oscillation in the MBH NAD$^+$ levels, which had a major peak at zeitgeber time (ZT) 18 (mid−dark phase) and a minor peak at ZT2 (early-light phase) and reached the lowest levels at ZT6−ZT10 (mid-late light phase) (Fig. 1a). This rhythmic fluctuation persisted after 2 weeks-exposure to continuous darkness (Fig. 1b), thus confirming a circadian rhythm of hypothalamic NAD$^+$ levels. NAD$^+$ assay in ARC punch biopsy samples also demonstrated a circadian oscillation with a nighttime elevation (Supplementary Fig. 1). Cosinor analysis revealed the rhythm parameters (mesor, amplitude, and period) of hypothalamic NAD$^+$ oscillations (Supplementary Table 1).

In mammalian cells, cellular NAD$^+$ levels largely depend on the iNAMPT activity[12] and also can be affected by blood eNAMPT in the cells having low iNAMPT expression levels such as neurons and pancreatic β-cells[14]. We thus examined whether the circadian rhythm of hypothalamic NAD$^+$ levels is in sync with the levels of iNAMPT in the MBH or eNAMPT in the plasma. NAMPT immunoblotting revealed distinct but opposite patterns of circadian fluctuations in the MBH iNAMPT and plasma eNAMPT expression levels (Fig. 1c, d). The MBH iNAMPT expression peaked in the early light phase (ZT2) and showed a trough in the early dark phase (ZT14) (Fig. 1c and Supplementary Table 1). In contrast, the plasma eNAMPT expression elevated during the dark period and decreased during the light period (Fig. 1d and Supplementary Table 1). These findings led us to speculate that a rise in MBH NAD$^+$ levels during the dark period may be related to the nocturnal rise in plasma eNAMPT.

Blood eNAMPT can be derived from adipocytes, hepatocytes, and leukocytes[13,15,30,31]. We thus examined the circadian fluctuations in the iNAMPT expression in epididymal white adipose tissues, the liver and leukocytes. Consistent with the previous report[27], hepatic iNAMPT protein expression showed circadian oscillations with a major elevation during the dark phase (Fig. 1e). This pattern of oscillation matched that of plasma eNAMPT (Fig. 1d, e). In contrast, circadian oscillation of iNAMPT expression in the epididymal adipose tissue showed the opposite pattern of circadian oscillation to those of blood eNAMPT and liver iNAMPT (Fig. 1d−f and Supplementary Table 1). In addition, a previous study has shown that eNAMPT secretion from adipocytes increases under fasting condition[15], arguing against the possible contribution of adipose tissue-derived eNAMPT to the elevation of blood eNAMPT levels during the physiological feeding period. As leukocytes secrete eNAMPT and may contribute to circulating eNAMPT concentrations[30], we isolated blood leukocytes to study circadian rhythms in leukocyte iNAMPT expression. We assayed *Nampt* mRNA expression levels in leukocytes due to little protein amount. Notably, leukocyte *Nampt* expression levels had a dramatic circadian oscillation with a peak at ZT10 (2 h before the beginning of the dark period) (Fig. 1g and Supplementary Table 1). Taken together, the nocturnal elevation of blood eNAMPT may relate to iNAMPT fluctuations in the liver or leukocytes.

The circadian rhythms of hormonal and metabolic parameters are strongly entrained by the light-dark cycle when food is plentiful, but can shift to match food availability when food supply is restricted[32]. We therefore tested whether food availability and light-dark cycle entrain the circadian oscillations in plasma eNAMPT level. Mice were either maintained on a reverse light/dark cycle and ad libitum-feeding (light-shift, light-dark entrainment) or allowed access to food only during the light period (ZT0−ZT12) under a normal light/dark cycle (food-shift,

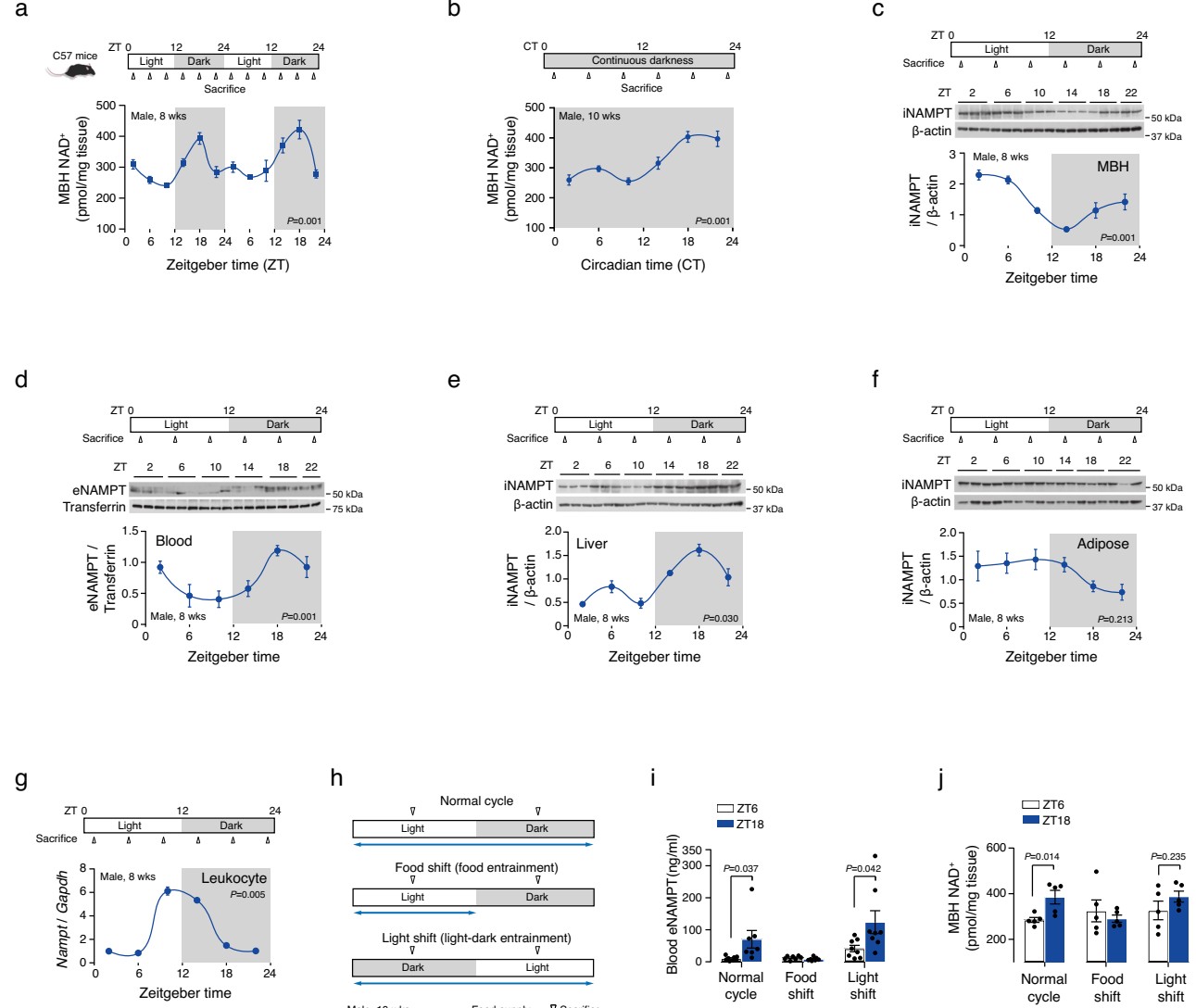

**Fig. 1 | Nocturnal rhythmic elevation of hypothalamic NAD$^+$ levels is in phase with that of plasma eNAMPT. a** Circadian fluctuation in the NAD$^+$ levels of the mediobasal hypothalamus (MBH) throughout two consecutive 12 h-light/dark cycles in 8-week-old mice ($n = 5$ mice). $P$ values was determined using JTK-cycle algorithm. **b** Circadian fluctuation in the MBH NAD$^+$ levels in mice after exposure to 2-week continuous darkness (CT18, $n = 4$; other time groups, $n = 5$ mice). $P$ values was determined using JTK-cycle algorithm. **c–f** Circadian oscillations in plasma eNAMPT (**d**) and the iNAMPT levels in the mediobasal hypothalamus (MBH, **c**), liver (**e**), and epididymal adipose tissues (**f**) in mice subjected to a 12-h light/dark cycle (plasma, MBH and liver ZT22, $n = 2$; adipose tissue ZT14, $n = 2$; other groups, $n = 3$

mice). $P$ values was determined using JTK-cycle algorithm. **g** Circadian rhythm of *Nampt* expression levels in leukocytes ($n = 3$ pooled samples, one sample pooled from three mice). $P$ values was determined using JTK-cycle algorithm. **h** Schematic diagram of the food and light shift study. **i, j** Altered circadian oscillations under light- and food-shifted condition in plasma eNAMPT levels (**i**) (Normal cycle ZT18 and Food shift ZT6/ZT18, $n = 7$; other groups, $n = 8$ mice) and MBH NAD$^+$ levels (**j**) ($n = 5$ mice). Unpaired *t*-test (two-sided). The shaded area in **a–g** represents the lights-off period. Two independent replicates were performed and measurements were taken from distinct samples. Samples derive from the same experiment and that gels/blots were processed in parallel. Data are presented as mean ± SEM.

food-entrainment) for 14 days (Fig. 1h). The circadian fluctuations in blood eNAMPT expression were blunted in the food-shift group but adjusted to shifted light-dark cycle (Fig. 1i). We also investigated whether altered food availability and light-dark cycle affected the circadian oscillations in hypothalamic NAD$^+$/iNAMPT levels. Circadian fluctuations in hypothalamic NAD$^+$ levels showed similar changes, although the nocturnal elevation in the light-shift group did not reach a statistical significance (Fig. 1j). The circadian rhythm of MBH iNAMPT expression was blunted by both manipulations, whereas hepatic iNAMPT expression showed a reverse in circadian rhythm by food shift and an adjustment to the shifted light-dark cycle (Supplementary Fig. 2). Therefore, light and food may act as important zeitgeber in the circadian fluctuations of hypothalamic NAD$^+$, plasma eNAMPT, and MBH/liver iNAMPT levels.

## Inhibition of systemic eNAMPT activity blocks nighttime increases in hypothalamic NAD$^+$ levels and locomotor activity

Metabolic parameters such as LMA, EE, and food intake (FI) are regulated by the hypothalamus and display marked circadian rhythms. Considering circadian oscillations in hypothalamic NAD$^+$ and plasma eNAMPT (Fig. 1a, b, d), we hypothesized that plasma eNAMPT may modulate the circadian rhythms of LMA, EE, or FI by affecting hypothalamic NAD$^+$ biosynthesis[15]. To test this hypothesis, we examined the effect of systemic eNAMPT inhibition on LMA, EE, and FI. eNAMPT inhibition was induced by continuous infusion of FK866, a chemical inhibitor of NAMPT, for 3 days via a peritoneum-implanted osmotic pump (Fig. 2a). Intraperitoneal (i.p.) infusion of FK866 (3.3 and 8 mg/kg) markedly blunted the nighttime increases in LMA and EE in a dose-dependent manner (Fig. 2b, c), whereas it did not block

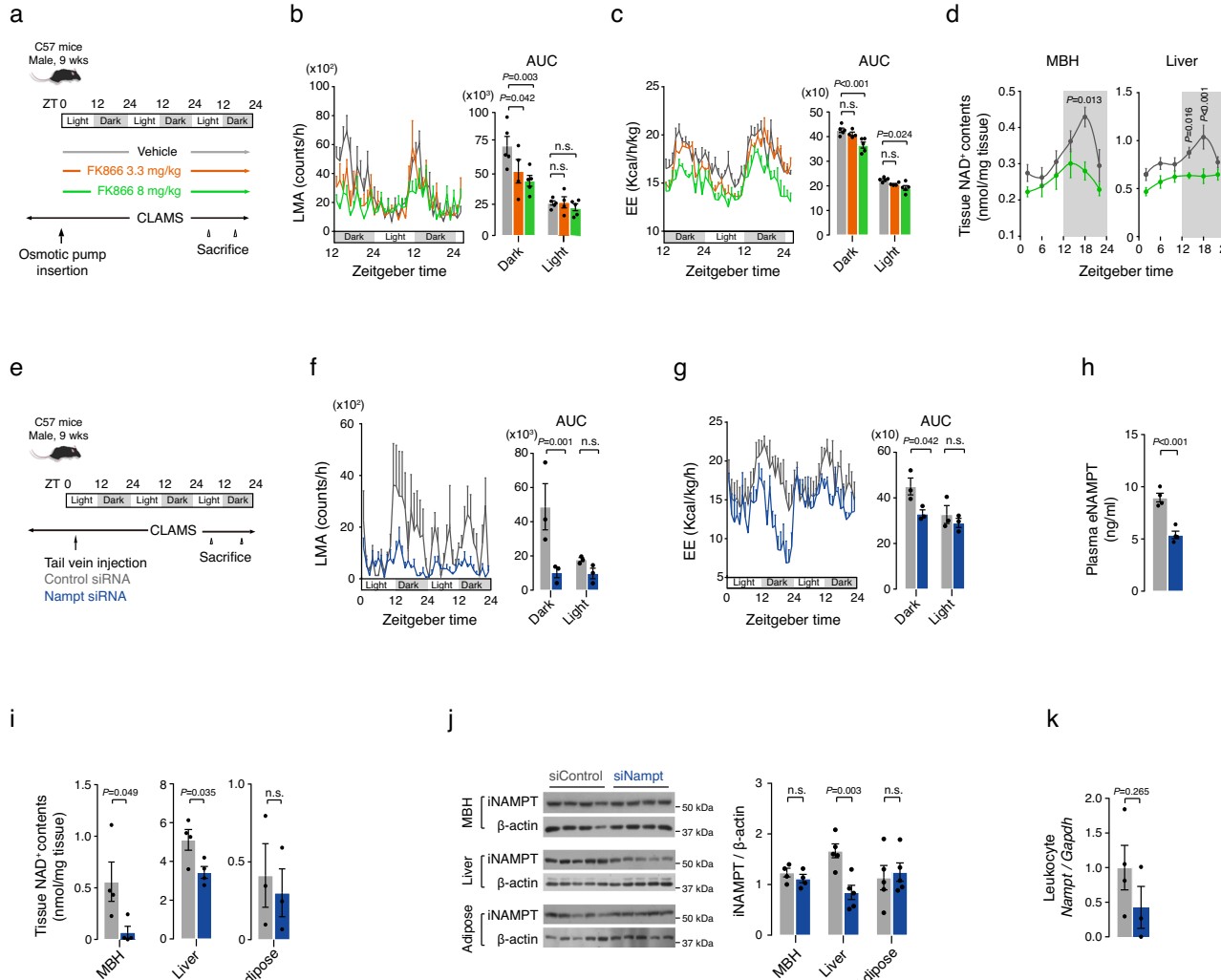

**Fig. 2 | Blockade of plasma eNAMPT activity abrogates circadian rhythms of locomotor activity and energy expenditure. a** Schematic diagram of the systemic FK866 study. ZT: zeitgeber time. **b**, **c** Systemic FK866 infusion-induced changes in circadian fluctuations in locomotor activity (LMA) (**b**) and energy expenditure (EE) (**c**) (Vehicle and FK866 8 mg/kg, $n = 5$; FK866 3 mg/kg, $n = 4$ mice). Two-way ANOVA followed by Fisher's Least Significance Difference (LSD) test. n.s.: not significant. **d** The mediobasal hypothalamus (MBH) and liver NAD$^+$ levels throughout a circadian cycle in mice that received an intraperitoneal (i.p.) infusion of either FK866 8 mg/kg or vehicle for 3 days ($n = 5$ mice at each time points). Repeated measures ANOVA followed by Fisher's LSD test. The shaded area in the figures represents the lights-off period. **e** Schematic diagram of the systemic Nampt siRNA (siNampt) study. **f**, **g** Effects of systemic administration of siNampt

on LMA (**f**) and EE (**g**) ($n = 3$ mice). Two-way ANOVA followed by Fisher's LSD test. n.s.: not significant. **h**, **i** Plasma eNAMPT concentrations and tissue NAD$^+$ levels in mice injected with siNampt or non-targeted control siRNA (siControl) at ZT18 on day 3 of treatment (adipose, $n = 3$; other groups, $n = 4$ mice). Unpaired $t$-test (two-sided). n.s.: not significant. **j**, **k** Changes in iNAMPT expression in the MBH, liver, adipose tissue (**j**), and leukocyte *Nampt* expression (**k**) on day 3 after systemic siNampt treatment (MBH, $n = 4$; Liver and Adipose, $n = 5$ mice; Leukocyte siControl, $n = 4$; Leukocyte siNampt, $n = 3$ pooled samples, one sample pooled from three mice). Unpaired $t$-test (two-sided). n.s.: not significant. Two independent replicates were performed for all studies and measurement were taken from distinct samples. Data are presented as mean ± SEM.

elevations in food intake during the dark period (Supplementary Fig. 3). Interestingly, nocturnal FI was enhanced by FK866 treatment, suggesting that nighttime elevation in systemic eNAMPT activity might relate to satiety formation.

FK866 infusion tended to lower the NAMPT enzyme activity in the plasma, liver, and leukocytes especially during the dark period, and lowered liver NAD$^+$ levels (Supplementary Fig. 4a–c and Fig. 2d). FK866 did not alter the MBH NAMPT activity throughout the circadian cycle (Supplementary Fig. 4d) but blocked a nocturnal rise in the MBH NAD$^+$ levels (Fig. 2d). This data suggests that systemic inhibition of NAMPT activity can disrupt normal circadian biosynthesis of NAD$^+$ in the hypothalamus.

We further tested the regulatory roles of eNAMPT on the circadian rhythms of LMA and EE with systemic administration of small inhibitory RNA targeting murine Nampt (siNampt) (Fig. 2e). Like FK866,

injection of siNampt (33 nmol/kg) into the tail vein abolished the nocturnal rises in LMA and EE and increased FI especially in the dark period (Fig. 2f, g and Supplementary Fig. 5). Therefore, siNampt-induced depletion of blood eNAMPT levels recapitulated the effects of chemical eNAMPT inhibition on LMA and EE.

Systemic siNampt treatment reduced plasma eNAMPT levels by ~40% and MBH NAD$^+$ levels by 80% compared with injection of non-targeting control siRNA (siControl) when measured at ZT18 (Fig. 2h, i). The liver NAD$^+$ levels were also mildly decreased while the adipose NAD$^+$ levels were not significantly affected by siNampt injection (Fig. 2i). Systemic administration of siNampt reduced the expression of liver iNAMPT while not significantly altering the iNAMPT protein expression levels in the MBH and epididymal fat (Fig. 2j). The leukocyte *Nampt* mRNA levels also tended to decrease in siNampt-injected mice (Fig. 2k). These findings suggest the possibility that liver- and/or

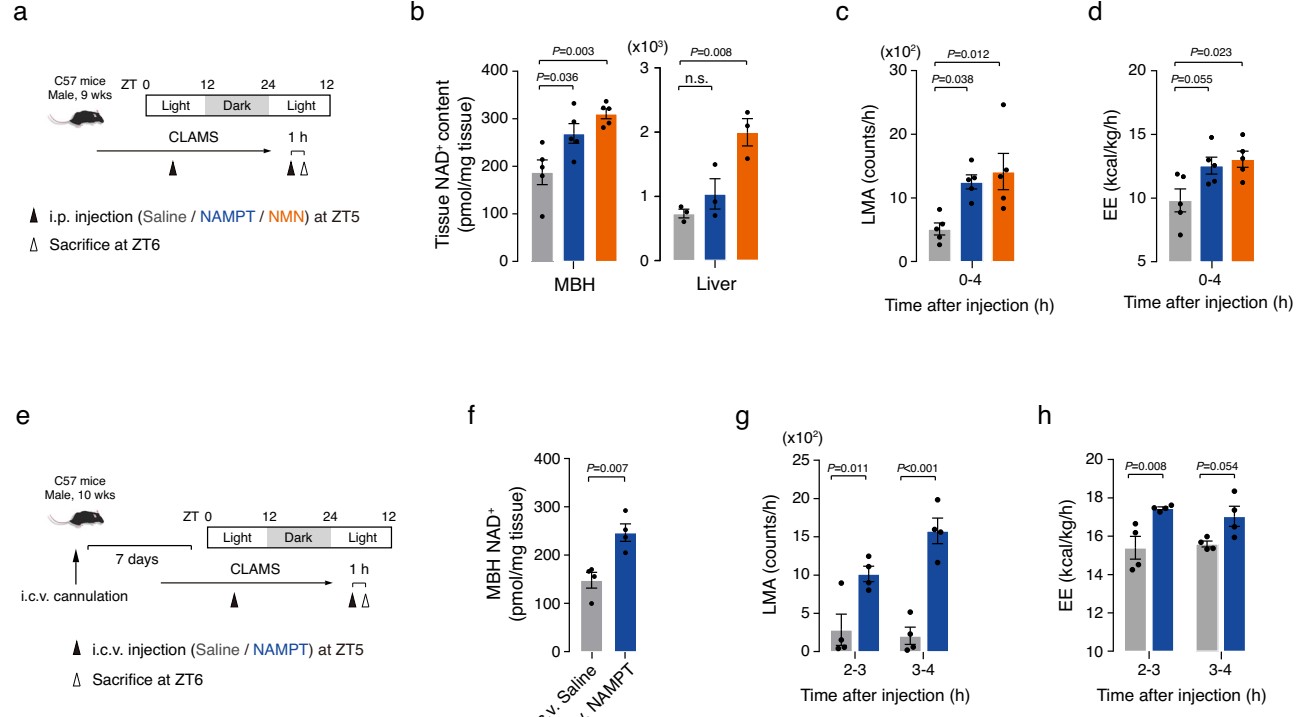

**Fig. 3 | Artificial elevation of systemic eNAMPT activity increases locomotor activity and energy expenditure. a** Schematic diagram of the intraperitoneal (i.p.) NAMPT/NMN injection study. ZT: zeitgeber time. **b** The mediobasal hypothalamus (MBH) and liver NAD$^+$ levels at ZT6 1 h after i.p. administration of NAMPT (0.3 mg/kg) and NMN (120 mg/kg) (MBH, $n$ = 5; Liver, $n$ = 3 mice). One-way ANOVA followed by Fisher's LSD test. n.s.: not significant. **c, d** Effects of i.p. administration of NAMPT and NMN on daytime locomotor activity (LMA) and energy expenditure

(EE) in mice ($n$ = 5 mice). LMA and EE were monitored for 4 h after injections. One-way ANOVA with Fisher's LSD test. **e** Schematic diagram of the intracerebroventricular (i.c.v.) NAMPT injection study. **f** The MBH NAD$^+$ levels at ZT6 1 h after i.c.v. NAMPT administration ($n$ = 4 mice). Unpaired $t$-test (two-sided). **g, h** Effects of i.c.v. administration of NAMPT (100 ng) on LMA (**g**) and EE (**h**) ($n$ = 4 mice). Two-way ANOVA followed by Fisher's LSD test. Two independent replicates were performed for all studies. Data are presented as mean ± SEM.

leukocyte-derived NAMPT may contribute to the dark-period elevation in plasma eNAMPT levels.

## Elevation of systemic eNAMPT activity stimulates hypothalamic NAD$^+$ levels and locomotor activity

We inversely tested the effects of artificial elevation of systemic eNAMPT activity on LMA, EE, and FI. To this end, we injected recombinant NAMPT protein intraperitoneally to normal mice at ZT5 when the endogenous plasma eNAMPT levels reach a trough (Figs. 1d, 3a). Before study, we confirmed the enzyme activity of NAMPT protein (Supplementary Fig. 6) and that i.p. injection of NAMPT (0.3 mg/kg) increased plasma eNAMPT levels at ZT6 to the level of endogenous eNAMPT at ZT18 (Supplementary Fig. 7). For comparison, we also injected NMN (120 mg/kg), a product of NAMPT enzymatic reaction.

During the light period, systemic administration of NAMPT and NMN significantly increased MBH NAD$^+$ levels at 1 h post-injection by 40% and 70%, respectively (Fig. 3b). NMN injection also elevated the liver NAD$^+$ levels (Fig. 3b). In contrast, NAMPT injection neither increased the hepatic NAD$^+$ levels nor the hepatic NAMPT activity and protein expression levels (Fig. 3b and Supplementary Fig. 8). We also administered the same doses of NAMPT protein and NMN during the dark period and examined the changes in the MBH and liver NAD$^+$ levels. The dark-period injection of NAMPT tended to increase MBH NAD$^+$ levels to a lesser degree compared with light-period injections (Supplementary Fig. 9). Similarly, NMN injection during the dark period modestly elevated the NAD$^+$ levels only in the liver (Supplementary Fig. 9). Therefore, the nighttime administration of NAMPT and NMN was less effective in terms of increasing the MBH or liver NAD$^+$ levels compared with daytime injections.

In contrast to FK866 and siNampt, i.p. injection of NAMPT and NMN stimulated LMA and EE during the light period (Fig. 3c, d). In addition, intracerebroventricular (i.c.v.) administration of NAMPT at a more than 100 times-lower dose (100 ng) increased LMA and EE as well as hypothalamic NAD$^+$ levels (Fig. 3e–h). These results suggest that increased eNAMPT activity can stimulate LMA and EE through a central mechanism. Artificial elevation in hypothalamic NAD$^+$ levels by eNAMPT and NMN administration in the light phase led to increased LMA and EE, which mimicked the physiological changes in LMA and EE during the dark period. Thus, these findings support the hypothesis that circadian fluctuation in blood eNAMPT and hypothalamic NAD$^+$ levels may contribute to the day–night rhythms of LMA and EE.

On the other hand, i.p.-injected NAMPT/NMN and i.c.v.-administered NAMPT suppressed fasting-induced hyperphagia (Supplementary Fig. 10). These data indicated that NAMPT/NMN-induced elevation in hypothalamic NAD$^+$ levels could lead to a negative energy balance through the stimulation of LMA/EE and suppression of FI. Overall, NAMPT and NMN administration induced the opposite changes to those of FK866 and siNampt.

## Blood eNAMPT acts on hypothalamic POMC and NPY/AgRP neurons

We studied the mechanism by which eNAMPT regulates LMA and EE in a circadian manner. We first tested the possibility that eNAMPT regulates the SCN clock activity. The SCN clock activity was assessed by bioluminescence monitoring of the PER2:luciferase activity in SCN-containing brain slices that were obtained from PER2:LUC mice (Fig. 4a). Treatment of recombinant NAMPT protein (100 nM) for 4 days did not significantly alter the period and amplitude of circadian

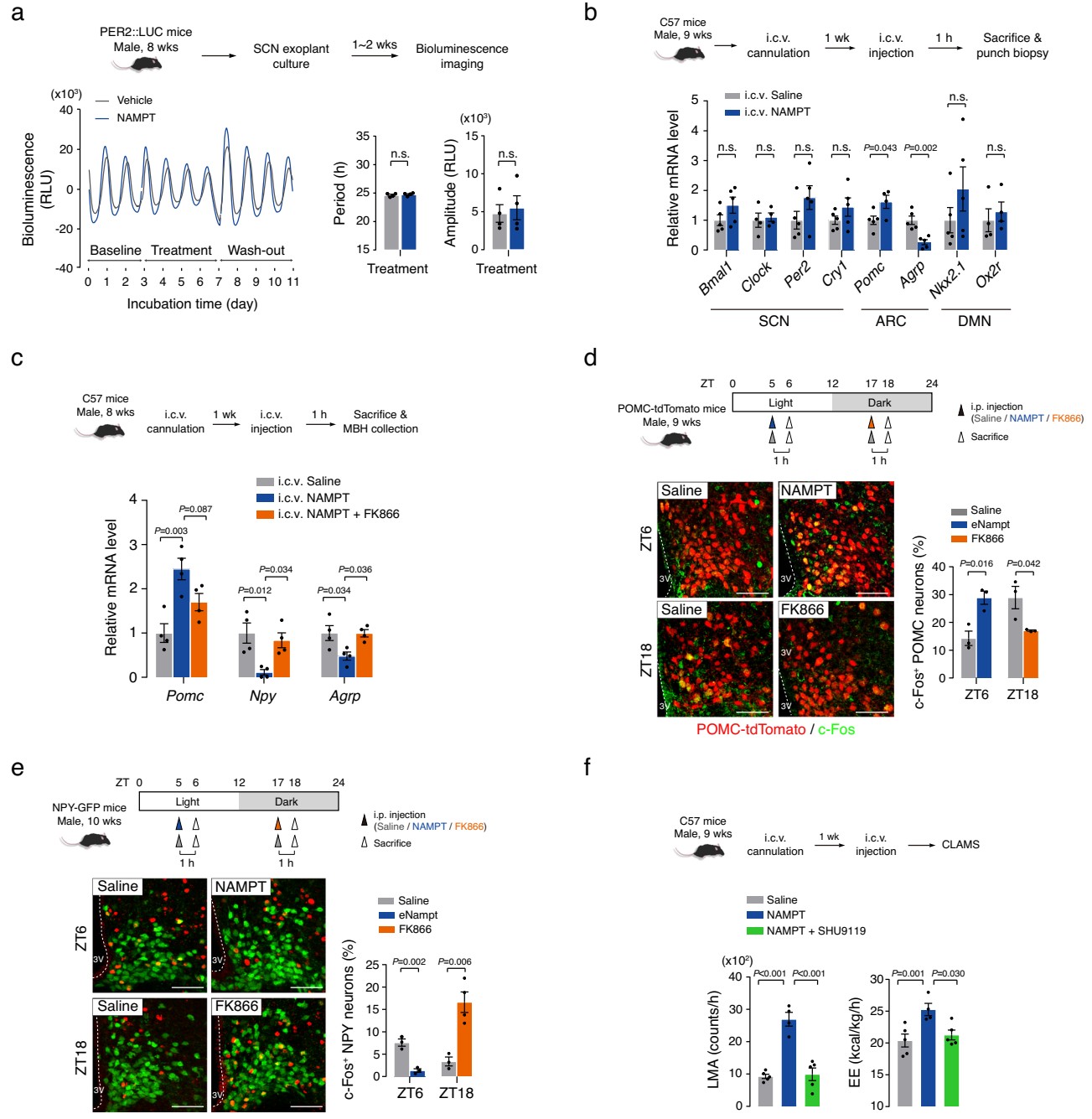

**Fig. 4 | Systemic eNAMPT activity regulates hypothalamic POMC and NPY/ AGRP neurons. a** Effects of NAMPT treatment on the SCN clock activity in SCN organotypic slice cultures obtained from PER2:LUC mice. SCN clock activity was determined by bioluminescence monitoring of PER2:luciferase activity. Representative circadian fluctuation curve are presented. The average values of period and amplitude of fluctuations were compared and plotted (*n* = 4 brain slices). One brain slice was obtained from one mice. Unpaired *t*-test (two-sided). n.s.: not significant. **b** Effects of intracerebroventricular (i.c.v.) administration of NAMPT protein (100 ng) on hypothalamic mRNA levels of genes. NAMPT was administered at ZT5, which was 1 h before sacrifice (*Clock, Pomc* NAMPT, *Ox2r, n* = 4; other groups, *n* = 5 mice). Unpaired *t*-test (two-sided). n.s.: not significant. SCN: suprachiasmatic nucleus, ARC: arcuate nucleus, DMN: dorsomedial nucleus. **c** Inhibition of the

NAMPT (100 ng)-induced changes in hypothalamic *Pomc, Npy*, and *Agrp* mRNA levels by FK866 (100 ng) pretreatment (*n* = 4 mice). One-way ANOVA followed by Fisher's LSD test. **d**, **e** POMC and NPY/AgRP neuronal activities measured with c-Fos expression at ZT6 and ZT18 after i.p. administration of saline or NAMPT (1 mg/kg) at ZT5 or saline or FK866 (10 mg/kg) at ZT17 (NPY c-Fos FK866, *n* = 4; other groups, *n* = 3 mice). The percentages of c-Fos⁺ POMC and NPY/AgRP neurons are presented with representative staining images. Unpaired *t*-test (two-sided) at ZT6 or ZT18. Scale bar = 50 μm. 3 V: third cerebroventricle. **f** Effects of i.c.v. administration of NAMPT (100 ng) alone or with SHU9119 (1 μg) on LMA and EE (Saline and NAMPT + SHU9119, *n* = 5; NAMPT alone, *n* = 4 mice). One-way ANOVA followed by Fisher's LSD test. Two independent replicates were performed for all studies. Data are presented as mean ± SEM.

oscillations of PER2:LUC activity when compared with those of vehicle-treated controls (Fig. 4a). In addition, i.c.v. administration of NAMPT did not significantly alter the mRNA expression of SCN clock genes (*Bmal1, Clock, Period 2, and Cryptochrome 1*) (Fig. 4b). These results collectively

suggest that systemic eNAMPT activity does not significantly affect the SCN clock activity. In contrast, our preliminary study showed the clock activity in SCN slices was profoundly suppressed when cultured in an FK866-containing medium (Supplementary Fig. 11), supporting the

importance of hypothalamic neuron iNAMPT in the SCN clock activity as previously suggested[33].

We next examined the expression of LMA/EE-regulating neuropeptides (proopiomelanocortin [Pomc], Agouti-related peptide [Agrp], neuropeptide Y [Npy]) in the ARC. The i.c.v. administration of NAMPT increased the Pomc expression but decreased the Agrp and Npy expressions in the ARC (Fig. 4b). Consistently, FK866 co-administration blocked the i.c.v. NAMPT-induced changes in Pomc, Npy, and Agrp transcript levels (Fig. 4c). These data imply that eNAMPT can modulate the hypothalamic Pomc, Npy, and Agrp expression via the mechanism dependent on the enzymatic activity of NAMPT. In contrast, i.c.v. NAMPT did not significantly alter the expression levels of SIRT1 target genes, Nkx2.1 and orexin receptor 2 (Ox2r)[34] in the hypothalamic dorsomedial nucleus (DMN) (Fig. 4b).

We also examined the effects of systemic eNAMPT modulation on the POMC and NPY/AgRP neuronal activity. The POMC neuronal activity was assessed by c-Fos expression in tdTomato-labeled POMC neurons at midday (ZT6) and midnight (ZT18) (Fig. 4d). The percentages of c-Fos⁺ POMC neurons were significantly different between ZT6 and ZT18 (13% vs. 28%, $P = 0.04$), indicating a significant circadian variation in the POMC neuronal activity according to the circadian fasting (ZT6)−feeding (ZT18) cycle (Fig. 4d). The low POMC neuronal activity at ZT6 was increased by i.p. injection of NAMPT (1 mg/kg), while the high POMC neuronal activity at ZT18 was suppressed by FK866 (10 mg/kg) injection (Fig. 4d). These changes in POMC neuronal activity were consistent with the changes in hypothalamic NAD⁺ levels (i.e., decrease by FK866 and increase by NAMPT) (Figs. 2d, 3b).

The NPY/AgRP neuronal activity, which was assessed through c-Fos expression in GFP-labeled NPY neurons, showed opposite changes to those of POMC neurons during circadian cycles and following NAMPT and FK866 injections (Fig. 4e). These data demonstrate that systemic eNAMPT activity can drive the circadian changes in POMC and NPY/AgRP neuronal activity.

We further tested the role of POMC and NPY/AgRP neurons in the eNAMPT-mediated regulation of LMA and EE. As POMC end product α-melanocyte-stimulating hormone (α-MSH) and AgRP are each endogenous agonist and antagonist of melanocortin 3 and 4 receptors (MC3/4R)[35], POMC/AgRP neurons are thought to regulate LMA and EE via these receptors[36–38]. We thus tested if the blockade of MC3/4 R with SHU9119 can inhibit the stimulatory effects of NAMPT on LMA and EE. Prior i.c.v. injection with SHU9119 prevented the NAMPT-induced changes in those metabolic parameters (Fig. 4f). These findings imply that eNAMPT controls LMA and EE via the central melanocortin system (i.e., POMC and AgRP neurons and the MC3/4R).

## Hypothalamic SIRT2 is a downstream mediator of eNAMPT actions

NAD⁺-dependent deacetylase SIRT1 is essential for the normal functions of POMC and NPY/AgRP neurons[21,22,39]. Mice lacking SIRT1 in POMC neurons are vulnerable to diet-induced obesity due to reduced EE[22], and SIRT1 deletion in NPY/AgRP neurons leads to reduced food intake and lower body mass[21]. We therefore examined the circadian oscillations of hypothalamic SIRT1 expression, and immunoblotting of MBH tissue blocks revealed a marked circadian rhythm in SIRT1 expression that peaked at ZT6−10 and decreased at ZT18 (Fig. 5a and Supplementary Table 2). The opposite circadian oscillation patterns of MBH SIRT1 expression and blood eNAMPT (Figs. 1d, 5a and Supplementary Tables 1, 2) led us to search for another sirtuin as a downstream mediator of eNAMPT, especially during the dark period. In contrast to SIRT1 expression, MBH SIRT2 expression remained stable throughout the day-night cycles (Fig. 5a and Supplementary Table 2).

Double staining of Sirt2/POMC (β-endorphin) or Sirt2/NPY-GFP confirmed Sirt2 expression in both types of neurons (Fig. 5b), and in vitro promoter assay revealed that NAMPT and NMN treatment increased the POMC transcriptional activity but repressed the AGRP

transcriptional activity in SH-SY5Y cells (Fig. 5c). Those effects were completely blunted by SIRT2 depletion (Fig. 5c and Supplementary Fig. 12a), thus demonstrating the importance of SIRT2 in the regulation of POMC and AGRP transcription by eNAMPT.

We further tested the role of SIRT2 in vivo by inhibiting hypothalamic SIRT2 through bilateral intra-MBH administration of siSirt2, which successfully depleted MBH Sirt2 in mice (Fig. 5d and Supplementary Fig. 12b). The depletion of Sirt2 in the MBH significantly blunted the NAMPT-induced changes in LMA, EE, and FI as well as those in the Pomc, Npy, and Agrp transcript levels (Fig. 5e, f and Supplementary Fig. 13). These findings suggested that hypothalamic SIRT2 acts as a downstream mediator for the role of eNAMPT in the modulation of LMA, EE, FI, and the expression of Pomc, Npy, and Agrp.

Given an importance of SIRT1 in normal functions of POMC and NPY/AgRP neurons[21,22,39], we additionally tested the involvement of SIRT1 in the eNAMPT regulation of POMC and AGRP gene transcription. In vitro promoter assay showed that SIRT1 knockdown markedly blocked the effects of NAMPT treatment on POMC and AGRP transcriptional activity (Supplementary Fig. 14). These findings suggest that in addition to SIRT2, SIRT1 may also mediate eNAMPT actions in the hypothalamus, and this possibility should be further tested in future studies.

## SIRT2 inhibits hypothalamic FOXO1 through deacetylation

We next sought to delineate the molecular mechanism by which SIRT2 regulates the transcription of POMC and AGRP. Sirtuins regulate transcription by deacetylating and regulating the activity of transcriptional factors such as FOXO1, p53, and nuclear factor of activated T-cells (NFAT)[18,39–41]. Among them, FOXO1 is well known as the regulator of both POMC and AGRP transcription[42,43], and previous studies have shown a direct interaction between SIRT2 and FOXO1 in adipocytes[44,45]. Sequential immunoprecipitation-immunoblotting of MBH protein extracts demonstrated the molecular interaction between FOXO1 and SIRT2 in the hypothalamus (Fig. 6a). Moreover, in N1 hypothalamic neuronal cells, Sirt2 overexpression decreased the acetylated form of FOXO1 (Fig. 6b). These data show that FOXO1 is a target of SIRT2 deacetylation in the hypothalamic neurons.

Next, we examined the functional interaction of SIRT2 and FOXO1 in the regulation of POMC and AGRP transcription. In line with previous reports[43], FOXO1 overexpression suppressed POMC transcription and stimulated AGRP transcription (Fig. 6c). SIRT2 alone expression increased POMC promoter activity but did not significantly alter AGRP promotor activity (Fig. 6c). Notably, SIRT2 coexpression inhibited the FOXO1's regulation of POMC and AGRP transcription (Fig. 6c), indicating antagonizing effects of SIRT2 on FOXO1 actions. SIRT1 overexpression recapitulated the effect of SIRT2 on FOXO1 regulation of POMC and AGRP transcription (Supplementary Fig. 15) and thus, both SIRT1 and SIRT2 may inhibit the transcriptional activity of FOXO1 in POMC and NPY/AgRP neurons.

The acetylation state of FOXO1 is known to affect its DNA binding, transcriptional activity, and ubiquitin-mediated degradation[46–48]. We thus tested if FOXO1 deacetylation alters the ability of FOXO1 to regulate the POMC and AGRP transcription using FOXO1 deacetylation mutant (6KR). In this mutant, six lysine residues corresponding to the proposed FOXO1 acetylation sites (K242, K245, K259, K262, K271, and K291) were replaced with arginine to prevent acetylation[46]. We confirmed a reduction in FOXO1 acetylation in cells expressing FOXO1-6KR (Supplementary Fig. 16a), and the expression of FOXO1-6KR, that mimicked SIRT2-induced FOXO1 deacetylation[44], failed to regulate AGRP and POMC transcriptional activity (Fig. 6d). Moreover, in vitro expression of FOXO1-6KR led to decreased FOXO1 protein levels (Supplementary Fig. 16b), indicating that FOXO1 deacetylation promotes the degradation of FOXO1, as shown previously[46,48]. Collectively, these findings suggest that SIRT2-mediated FOXO1 deacetylation

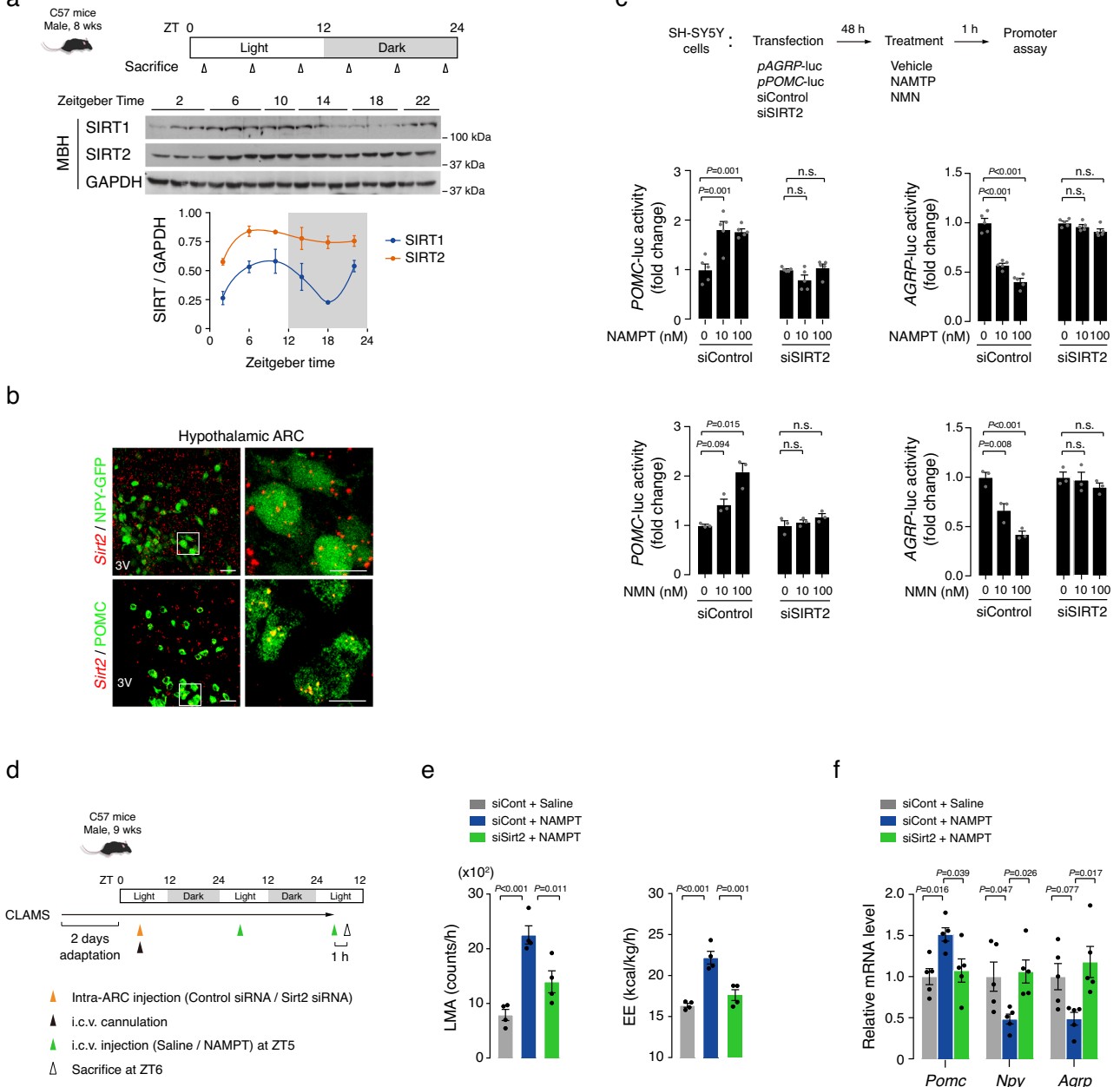

**Fig. 5 | SIRT2 mediates the hypothalamic actions of eNAMPT. a** Circadian patterns of the expression of SIRT1 and SIRT2 in the MBH of normal mice (ZT10, 22, $n = 2$; other groups, $n = 3$ mice). Two independent replicates were performed. ZT zeitgeber time. **b** Representative images of *Sirt2* fluorescent in situ hybridization and POMC (ß-endorphin) or NPY-GFP immunofluorescence double staining showing *Sirt2* expression in the POMC and NPY/AgRP neurons of the hypothalamic arcuate nucleus (ARC). Two independent replicates were performed. Scale bars = 25 µm and 10 µm (magnified images). **c** Effects of NAMPT/NMN treatment on *POMC* and *AGRP* promoter activities in SH-SY5Y cells with or without *SIRT2* knockdown (NAMPT, $n = 5$; MNM, $n = 3$ cell wells). One-way ANOVA followed by Fisher's LSD test. n.s.: not significant. Three independent experiments were performed to validate the results. **d** Schematic diagram of the hypothalamic Sirt2 depletion study. Some animals were subjected to killing study without CLAMS. **e, f** Effect of hypothalamic *Sirt2* depletion with bilateral ARC injection of siSirt2 on i.c.v. NAMPT (100 ng)-induced changes in LMA, EE, and hypothalamic neuropeptide expression in mice (LMA/EE, $n = 4$; neuropeptide, $n = 5$ mice). One-way ANOVA followed by Fisher's LSD test. Two independent replicates were performed. Data are presented as mean ± SEM.

represses the expression and functions of FOXO1 in hypothalamic neuron cells.

We further examined the importance of FOXO1 in the hypothalamic actions of eNAMPT in mice. I.c.v. NAMPT administration significantly reduced the total FOXO1 levels in the MBH (Fig. 6e). Prior i.c.v. injection of proteasome inhibitor MG132 attenuated the NAMPT-induced reduction in hypothalamic FOXO1 levels (Supplementary Fig. 17). Furthermore, the circadian changes in hypothalamic FOXO1 displayed an inverse relationship with circulating eNAMPT

levels in normal mice (Figs. 1d, 6f). Thus, this in vivo evidence suggest that eNAMPT may promote hypothalamic FOXO1 degradation, which may occur through NAD+/SIRT-mediated FOXO1 deacetylation[46,48]. Bilateral intra-ARC injection of siSirt2 significantly blunted the i.c.v. NAMPT-induced decrease in hypothalamic FOXO1 levels at ZT6 (Fig. 6g). Together, these results support an important role of SIRT2 in eNAMPT-induced FOXO1 degradation.

We finally tested whether hypothalamic *Foxo1* overexpression can block eNAMPT actions by injecting either Foxo1-GFP-expressing

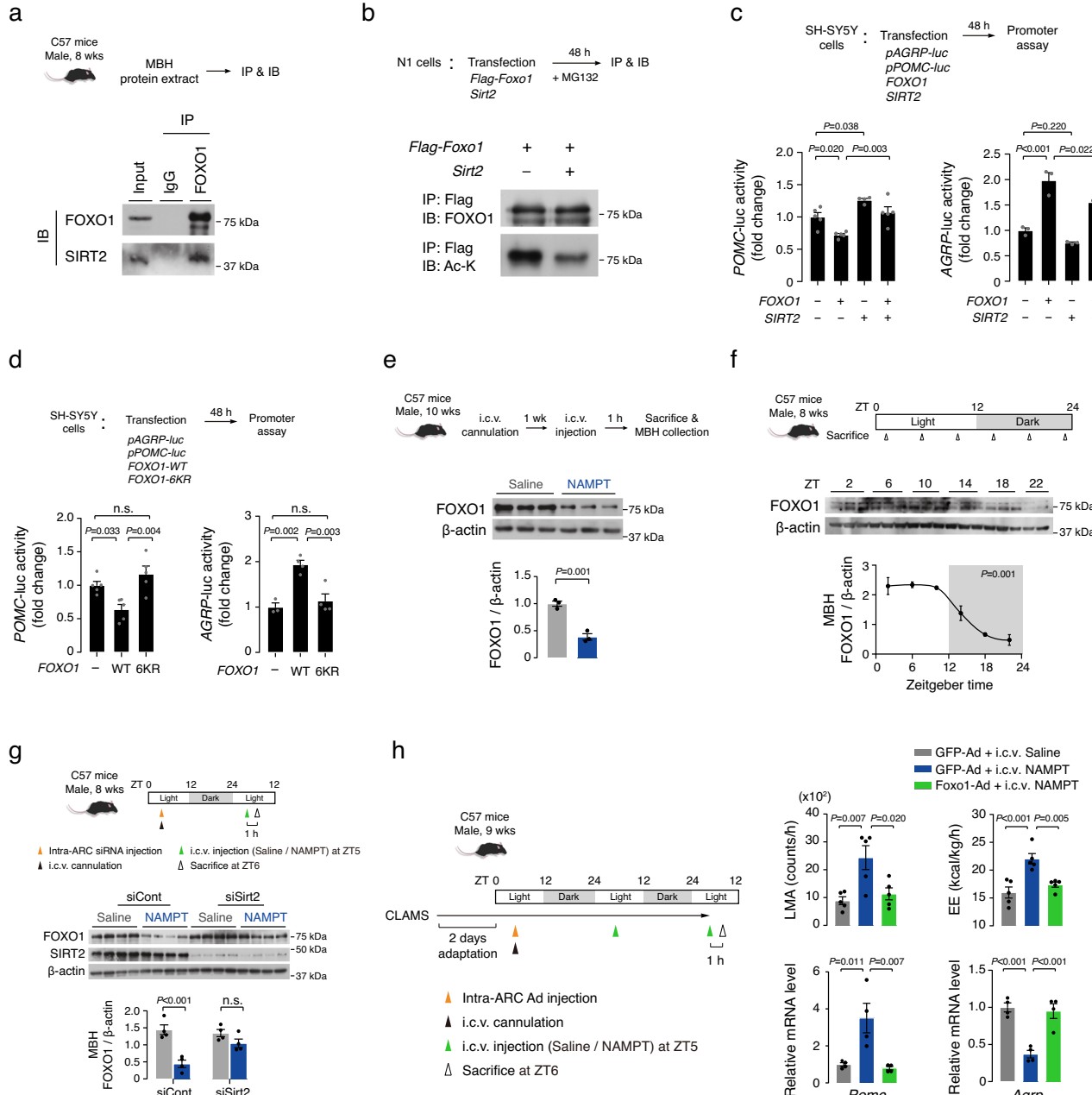

**Fig. 6 | eNAMPT inhibits hypothalamic FOXO1 via SIRT2−mediated deacetylation. a** Molecular interaction of SIRT2 and FOXO1 in C57 mouse hypothalamic protein extracts. IP: immunoprecipitation, IB immunoblotting. **b** Effect of *Sirt2* overexpression on FOXO1 acetylation in hypothalamic N1 cells. Ac-K: acetylated lysine. **a**, **b** Two independent replicates were performed. **c** Antagonizing effect of *SIRT2* overexpression on *FOXO1* overexpression-induced changes in the *POMC* and *AGRP* promoter activities (*POMC SIRT2* alone, *n* = 4; *POMC* other groups, *n* = 5; *AGRP* all groups, *n* = 3 cell wells). One-way ANOVA followed by Fisher's LSD test. **d** Comparison of the effect of FOXO1 wild type (WT) and deacetylation mutant (6KR) overexpression in the regulatory ability of *POMC* and *AGRP* transcription (*POMC*, *n* = 5; *AGRP* Vector, *n* = 3; *AGRP* WT and 6KR, *n* = 4 cell wells). One-way ANOVA followed by Fisher's LSD test. n.s. not significant. **c**, **d** Three independent replicates were performed. **e** Effect of i.c.v. administration of NAMPT (100 ng) on hypothalamic FOXO1 expression (*n* = 3 mice). Unpaired *t*-test (two-sided). **f** Circadian rhythms of hypothalamic FOXO1 expression levels in normal mice (ZT22, *n* = 2; other groups, *n* = 3 mice). *P* value was determined using JTK-cycle algorithm. The shaded area in the figures represents the lights-off period. **g** Effect of hypothalamic *Sirt2* knockdown on i.c.v. NAMPT-induced decreases in hypothalamic FOXO1 expression (*n* = 4 mice). One-way ANOVA followed by Fisher's LSD test. n.s. not significant. **h** Effect of hypothalamic *Foxo1* overexpression by Foxo1-Ad injection on i.c.v. NAMPT (100 ng)-induced changes on LMA, EE, and hypothalamic *Pomc* and *Agrp* expression (EE and LMA, *n* = 5; *Pomc* and *Agrp*, *n* = 4 mice). One-way ANOVA followed by Fisher's LSD test. **e-h** Two independent experiments were performed to validate the results. Data represent the mean ± SEM.

adenovirus (Foxo1-Ad) or GFP alone-expressing control virus (GFP-Ad) into bilateral ARC prior to i.c.v. NAMPT injection (Fig. 6h). We confirmed successful virus injection in the ARC and increased MBH *Foxo1* expression (Supplementary Fig. 18). Overexpression of *Foxo1* alone in the ARC decreased EE, and hypothalamic *Pomc* expression levels and tended to lower LMA, but increased *Agrp* expression levels (Supplementary Fig. 19). Moreover, *Foxo1* overexpression blunted NAMPT-induced changes in LMA and EE as well as in *Pomc* and *Agrp* expression (Fig. 6h). These data indicate that suppression of hypothalamic FOXO1 is critical for the actions of eNAMPT.

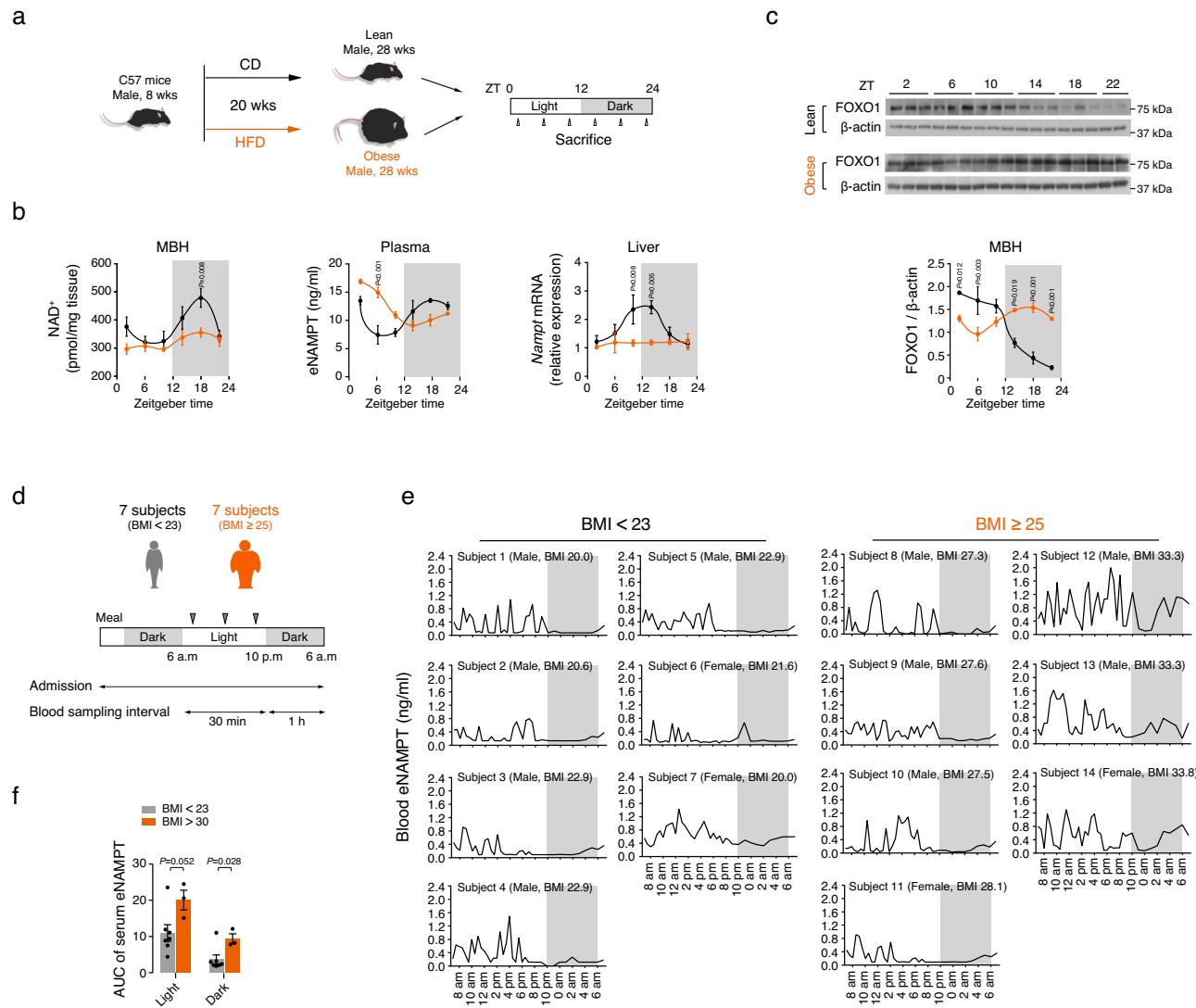

**Fig. 7 | Obesity disrupts the circadian rhythms of the blood eNAMPT−hypothalamic NAD⁺−FOXO1 axis. a** Schematic diagram of the obese mice study. **b** Disrupted circadian rhythms in hypothalamic NAD⁺ levels, plasma eNAMPT, and liver *Nampt* in obese mice fed a high-fat diet for 20 weeks (Lean, *n* = 3; Obese, *n* = 5 mice). Repeated measures ANOVA followed by Fisher's LSD test. Two independent experiments were performed to validate the results. **c** Comparison of circadian rhythms in hypothalamic FOXO1 expression levels in lean and 20 weeks-HFD-fed obese mice (ZT22, *n* = 2; other groups, *n* = 3 mice). Repeated measures ANOVA followed by Fisher's LSD test. Two independent experiments were performed to validate the results. Samples derive from the same experiment and that gels/blots were processed in parallel. **d** Schematic diagram of the human eNAMPT study. **e** The 24-h profiles of serum eNAMPT concentrations in young healthy volunteers [7 lean subjects with a body mass index (BMI) < 23 and 7 subjects with obesity, their BMI ≥ 25]. Note the lack of oscillation during the light-off period in lean subjects but the presence of significant oscillations in subjects with BMI > 30. **f** The area under the curve (AUC) of serum eNAMPT during light and dark periods in lean subjects with BMI < 23 and subjects with obesity (BMI > 30) (BMI < 23, *n* = 7, BMI > 30, *n* = 3). Unpaired *t*-test (two-sided). The shaded area in **b**, **c**, **e** represents the lights-off period. All data represent the mean ± SEM.

## Obesity disrupts circadian fluctuations in plasma eNAMPT−hypothalamic NAD⁺−FOXO1 axis

Obesity is strongly associated with disruptions in central and peripheral molecular clockworks[49]. We thus examined whether circadian oscillations in liver iNAMPT−plasma eNAMPT−hypothalamic NAD⁺ axis may be altered by obesity in mice. Obesity was induced by the consumption of HFD for 20 weeks (Fig. 7a). Mice with diet-induced obesity (DIO) had disrupted circadian oscillation in MBH NAD⁺ levels, plasma eNAMPT, and liver *Nampt* expression levels (Fig. 7b and Supplementary Table 3). The circadian fluctuation in hypothalamic FOXO1 expression levels was reversed and that of LMA was blunted in obese mice as well (Fig. 7c, Supplementary Fig. 20 and Supplementary Table 3).

A previous human study reported a distinct circadian rhythm in plasma eNAMPT/visfatin with a peak in the afternoon, which corresponds to midnight in rodents, and a dip in the early morning[50]. We thus tested whether obesity may alter the eNAMPT circadian rhythm in humans. We examined the 24-h profiles of blood eNAMPT levels in 14 healthy young (≤ 30 years-old) volunteers (7 males, 7 females) with no medical history of chronic illness or malignancy, of whom 7 were lean subjects (BMI < 23 kg/m²) and 7 were individuals with obesity (BMI ≥ 25 kg/m²). The characteristics of the subjects are shown in Supplementary Table 4. Blood samples were collected every 30 min from 6 a.m. to 10 p.m., and then hourly until 6 a.m. in the next morning (Fig. 7d). During the study, three meals were provided at 8 a.m., 12:30 p.m., and 6 p.m., and lights were turned on at 6 a.m. and turned off at 10 p.m. Frequent blood sampling showed a pronounced oscillation pattern of serum eNAMPT concentrations in lean young subjects that had 3 to 7 pulsatile oscillations during the light-on period but no definite oscillations during the light-off period (Fig. 7e).

Although a distinct difference in the eNAMPT oscillation between the light-on period and light-off period, the timing of daytime peaks varied among individuals. Notably, 3 subjects (2 males and 1 female) with a BMI > 30 kg/m² showed significant oscillations during the light-off period, whereas subjects with mild obesity (BMI 25–29.9 kg/m²) had no oscillation during this period (Fig. 7e). The area under the curve (AUC) value of serum eNAMPT during the dark period was significantly higher in the subjects with obesity (BMI > 30) than in the lean subjects, although the AUC of serum eNAMPT during the light period also showed a tendency to increase in individuals with obesity (Fig. 7f). Although our data was obtained from a small number of young healthy subjects, these human data suggest that day-night difference in blood eNAMPT oscillations can be altered under chronic overnutrition conditions and this phenomenon might relate to altered day-night rhythms of physical activity in individuals with obesity[51,52].

## Discussion

Our findings are the first to demonstrate a 24-h cyclic oscillation in the hypothalamic NAD⁺ levels with a major peak around midnight. This rhythm persists under continuous darkness and is profoundly disturbed by the reversal of the light-dark phase and daytime-restricted feeding. Notably, the nocturnal elevation in hypothalamic NAD⁺ levels seems to relate to circulating eNAMPT considering the low hypothalamic iNAMPT expression and enzyme activity during this period. Supporting this possibility, eNAMPT has a high enzymatic activity of NAMPT[14] and its blood levels rise during the night. Moreover, systemic administration of the NAMPT inhibitor FK866 mitigates a nighttime elevation in hypothalamic NAD⁺ levels. These findings suggest that eNAMPT-mediated regulation of hypothalamic NAD⁺ biosynthesis occurs in a circadian manner.

A previous study has reported the circadian profiles of blood eNAMPT levels in 6- and 18-months-old mice[13]; in that study, blood eNAMPT levels showed a similar circadian fluctuation in 6-months-old mice, although the amplitude of fluctuation was less when compared with the 8–10 weeks-old mice used in our study. Indeed, the blood NAMPT level and its circadian fluctuations decrease with age[13]. Consistent with the findings from rodents, a previous study has shown that humans also display a circadian pattern of oscillation rhythm in blood eNAMPT levels with a peak in the early afternoon[50], which corresponds to midnight in rodents. Interestingly, frequent blood sampling in our human study revealed 3–7 pulsatile oscillations, a sort of ultradian rhythm, during the light-on period, although these oscillations were irregular in terms of frequency, period, and amplitude. Notably, there was a near-complete absence of oscillation during the light-off period and this day-night difference may represent a circadian pattern of fluctuation in blood eNAMPT concentrations.

There is compelling evidence that the eNAMPT upregulates hypothalamic NAD⁺ levels[13,15], but the detailed mechanisms of this regulation remain elusive. Circulating eNAMPT might directly act on the hypothalamus to promote NAD⁺ biosynthesis. The finding from a hypothalamic explant experiment demonstrated a direct eNAMPT action, thus supporting this possibility[15]. Alternatively, eNAMPT might produce NMN in the bloodstream through NAMPT enzyme reaction and provide NAD⁺ precursor to hypothalamic neurons[14]. Indeed, exogenous NMN is efficiently transported to the MBH and upregulates the hypothalamic NAD⁺ levels[53]. However, the occurrence of NAMPT reaction in circulation is debatable due to the low plasma concentrations of its critical components ATP and 5-phosphoribosyl 1-pyrophosphate[54]. Considering the enrichment of eNAMPT in blood extracellular vesicle (EVs)[13], NAMPT reaction might take place within blood EVs. To test this possibility, it will be worth to examine the presence or the amount of NAMPT reaction substrates (e.g., nicotinamide, 5-phosphoribosyl 1-pyrophosphate), cofactor (ATP), and product (NMN) in blood EVs. It is also possible that eNAMPT produces NAD⁺ in the bloodstream and blood NAD⁺ is transported to the extracellular space in the MBH and then enter neurons or glia to elevate intracellular NAD⁺ levels. We have previously shown that NAD supplement effectively elevates hypothalamic NAD⁺ levels through a connexin 43-dependent mechanism[55]. However, there is yet little evidence on the presence of blood NMNAT, which catalyzes NMN-to-NAD⁺ conversion. Aside from the mode of eNAMPT actions, it is unclear whether the hypothalamic NAD⁺ levels are altered intracellularly and/or extracellularly.

In terms of the specific organs or tissues responsible for the secretion of eNAMPT to drive its circadian rhythm, previous studies have shown that brown and white adipocytes release eNAMPT[15,53]. Moreover, adipocyte-specific Nampt overexpression and depletion altered blood eNAMPT levels and hypothalamic NAD⁺ levels in female mice[13,15]. It is therefore possible that adipocyte-secreted eNAMPT contributes to the circadian rhythms of blood eNAMPT, especially in females. However, eNAMPT secretion by adipocytes was enhanced by fasting[15] whilst blood eNAMPT levels were elevated during the physiological feeding period. Therefore, the contribution of adipocytes to nocturnal elevation in blood eNAMPT is less likely. On the other hand, liver iNAMPT levels showed a dramatic circadian oscillation with dark-period elevation. Second, the liver iNAMPT expression levels correlated well with those of plasma eNAMPT under normal day/night cycle as well as in food- and light-shift conditions. Third, systemic administration of siNampt significantly decreased both liver iNAMPT and plasma eNAMPT levels, while it did not significantly alter the adipose iNAMPT expression. These data collectively suggest the liver is a potential contributor to circadian oscillation in plasma eNAMPT expression. Future studies are needed to examine whether hepatocytes secrete eNAMPT in a circadian dependent manner. As the ability of adipocytes to secrete eNAMPT depended on SIRT1-mediated NAMPT deacetylation[15], it will be interesting to study if hepatocytes may secrete eNAMPT through sirtuin- and EV-dependent mechanisms as shown in adipocytes[13,15]. Indeed, immunoblotting assay of plasma eNAMPT revealed two bands, which may indicate post-transcriptional modification of eNAMPT. It has been suggested that leukocytes potentially contribute to plasma eNAMPT concentrations[30]. In our study, leukocyte Nampt expression and enzyme activity exhibited the circadian pattern with a dramatic rise just before or at the beginning of dark period, respectively, and they were repressed by FK866 or siNampt treatment; however, hematopoietic cell iNAMPT overexpression failed to increase plasma eNAMPT levels[56]. Hence, the contribution of leukocytes in the day/night oscillations in blood eNAMPT remains to be clarified.

Regarding physiological situations in which plasma eNAMPT stimulates physical activity, eNAMPT secretion by adipocytes is increased under fasting condition[15]. Moreover, depletion of adipocyte NAMPT reduced LMA in a 48-h-fasted condition[15]. Thus, adipocyte-derived eNAMPT may involve fasting-enhanced LMA. In contrast, hepatocytes or leukocytes might secrete eNAMPT in the dark period to stimulate LMA during the physiologically active dark period. The hypothesis of differential roles and cellular sources of blood eNAMPT according to physiological context will be worth testing in the future using tissue-specific Nampt knockout mice. Meanwhile, eNAMPT-mediated regulation of LMA appeared to be influenced by sex because adipocyte-specific NAMPT depletion reduced LMA in female mice but not in male mice[15]. As our animal studies were conducted in male mice, the role of plasma eNAMPT in the circadian regulation of LMA needs to be tested in female mice.

Consistent with LMA, the day/night rhythms of EE were also significantly affected by systemic modulation of eNAMPT. As LMA is a component of EE, these changes in EE can be explained by alterations in LMA. In addition, eNAMPT could affect the EE through modulation of other components of EE such as adipose tissue thermogenesis because eNAMPT target POMC and AgRP neurons regulate thermogenesis[57,58]. On the other hand, eNAMPT also appears to be

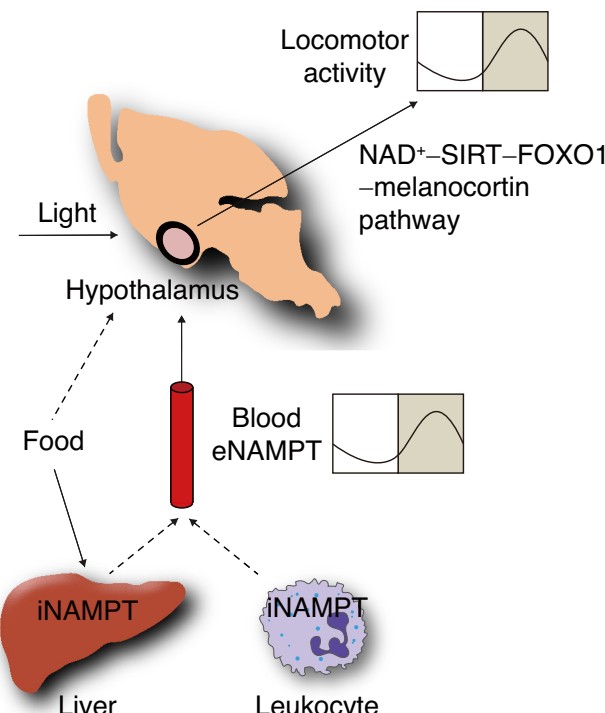

**Fig. 8 | Schematic illustration of a potential role for eNAMPT in hypothalamic circadian regulation of locomotor activity.** Light and food availability control eNAMPT secretion from the liver or leukocytes to systemic circulation in a circadian manner. Blood eNAMPT acts upon the hypothalamus to drive circadian rhythms in locomotor activity via hypothalamic NAD⁺–SIRT–FOXO1–melanocortin pathways.

involved in satiety formation in the middle of the dark phase, considering that systemic eNAMPT inhibition with FK866 and siNampt infusion increased food intake during the night. Conversely, an increase in plasma eNAMPT by NAMPT administration suppressed fasting-induced feeding and thus, it will be interesting to examine whether eNAMPT secreted by adipocytes may suppress appetite during a fast.

To decipher the mechanism underpinning hypothalamic eNAMPT actions, we first tested the ability of eNAMPT to alter the SCN PER2: luciferase activity and found null results. Moreover, the SCN expression levels of clock genes *Bmal1*, *Clock*, *Per2*, and *Cry1* were unaltered by NAMPT treatment, implying little contribution of systemic eNAMPT activity on the master clockwork. We also examined the DMH expression of SIRT1 target gene *Ox2r* and SIRT1's transcription partner *Nkx2.1*[34] and found no significant changes, at least in the context of acute NAMPT treatment. In contrast, i.p. injections of NAMPT or FK866 significantly altered the *Pomc*, *Npy*, and *Agrp* transcript levels. More importantly, the circadian fluctuations in POMC and NPY/AgRP neuronal activities were profoundly affected by the modulation of systemic eNAMPT activity. These findings suggest that eNAMPT may regulate LMA, EE, and FI through its actions on these neurons.

Sirtuins mediate many salutary roles of the NAMPT–NAD⁺ axis[11,16]. We found that hypothalamic SIRT2 may serve as an important downstream mediator of plasma eNAMPT–hypothalamic NAD⁺ axis. SIRT2 is expressed in ARC POMC and NPY/AgRP neurons. As for the downstream mediator of SIRT2, we focused on FOXO1, a well-known regulator of POMC and NPY/AgRP transcription. Indeed, FOXO1 binds the AgRP and NPY promoters to simulate their transcription while FOXO1 inhibits the signal transducer and activator of transcription 3 (STAT3)-stimulated POMC transcription[42,43]. We found that SIRT2 interacted with FOXO1 to suppress FOXO1 regulation of POMC and NPY/AGRP transcription. Furthermore, the depletion of SIRT2 and overexpression of FOXO1 blunted the hypothalamic actions of eNAMPT on LMA, EE

and *Pomc* and *Agrp* transcription in mice. These findings support the hypothesis that eNAMPT stimulates LMA and EE through SIRT2-mediated FOXO1 inhibition and the resulting functional changes in POMC and NPY/AgRP neurons. As for the molecular mechanisms of SIRT2 and FOXO1 interaction, we showed that SIRT2 deacetylated and then promoted FOXO1 degradation.

Similar to SIRT2, depletion of SIRT1 significantly blocked the effects of NAMPT treatment on POMC and AgRP transcription. Moreover, SIRT1 coexpression reversed FOXO1 overexpression-induced changes in POMC and AGRP transcription. Therefore, in addition to SIRT2, SIRT1 may have a role in the process of eNAMPT regulation of POMC and AGRP transcription. The expression of SIRT1 in the MBH showed a distinct circadian oscillation, with a rise in the light phase and a fall in the dark phase. The opposing circadian patterns of hypothalamic SIRT1 and blood eNAMPT may be against the role of SIRT1 as a major downstream mediator of eNAMPT. However, SIRT1 activity largely depends on the cellular NAD⁺ levels and the MBH NAD⁺ levels were elevated in the dark period. It is thus possible that during the dark period, the SIRT1 activity in the MBH neurons may maintain a certain level, which is required for mediating eNAMPT actions.

Body clockwork and obesity influence each other[3,59]. Therefore, the robustness of eNAMPT-mediated circadian regulation of LMA and hypothalamic NAD⁺ needs to be tested under obese conditions. A previous study reported that the hepatic and adipose protein levels of iNAMPT were markedly reduced in HFD-fed obese mice[53]. In addition, we found that DIO mice had disrupted circadian oscillations in the liver iNAMPT and plasma eNAMPT expression and MBH NAD⁺ levels. Moreover, DIO disrupted circadian rhythms in hypothalamic FOXO1 expression and LMA. In line with the data in rodents, three human subjects with obesity (a BMI over 30) showed a disrupted day-night pattern of blood eNAMPT, although these finding needs to be confirmed in a larger number of human subjects. Taken together, the eNAMPT-mediated regulatory axis can be readily disrupted in animals and humans with obesity. Similar to obesity, aging blunts the circadian oscillation in plasma eNAMPT in mice[13]. Notably, adipose tissue Nampt overexpression and systemic NMN administration rescued the aging-associated reduction in hypothalamic NAD⁺ content and nighttime LMA activity[13,53]. Therefore, obesity-induced disruptions in the blood eNAMPT–hypothalamic NAD⁺–LMA axis may be treatable using NAMPT and NMN.

In conclusion, our study reveals a novel mechanism of circadian physiology involving eNAMPT-mediated regulation of hypothalamic NAD⁺ biosynthesis (Fig. 8) and proposes a potential therapeutic strategy for obesity-related circadian disorders.

## Methods
### Materials
Recombinant human NAMPT protein was obtained from Adipogen International (AG-40A-0018). We purchased FK866 from Enzo Life Sciences (ALX270-501), SHU9119 from Phoenix Pharmaceuticals (043-24) and NMN from Sigma (1094-61-7).

### Animals
Male C57BL/6 mice (8–10-weeks old) were obtained from Orient Bio. Animals were housed under temperature-controlled conditions (22 ± 1 °C), humidity (55 ± 5%), and subjected to a 12-h light/dark cycle with lights on from 8 a.m. with ad libitum access to food to 8 p.m. unless indicated otherwise. Thus, 8 a.m. is zeitgeber time 0 (ZT0). Mice were fed a standard diet (Agripurina) with free access to food and water unless indicated otherwise. To induce diet-induced obesity, mice were fed a high-fat diet (60% fat; Research Diets, D12492). Mice with tdTomato-labeled POMC neurons were generated by mating POMC-Cre mice (provided by Dr. Joel K. Elmquist, University of Texas Southwestern Medical Center) with mice carrying the reporter Rosa26-loxP-STOP-loxP-tdTomato allele (Jackson Laboratory, 007909).

NPY-hrGFP mice (Jackson Laboratory, 006417) were used for the purpose of AGRP neuron labeling. Dr. Joshep S. Takahashi (University of Texas Southwestern Medical Center) provided PER2:LUC knock-in mice used in the SCN clock study. All procedures were approved by the Institutional Animal Care and Use Committee of the Asan Institute for Life Sciences under the project license 2012-11-024 (Seoul, Korea).

## Cell culture

SH-SY5Y human neuroblastoma cells (ATCC, CRL-2266) and murine hypothalamic N1 cells (Cedarlane, CLU101) were maintained in Dulbecco's modified Eagle's medium (DMEM) containing 10% fetal calf serum and penicillin and streptomycin (100 units/ml each).

## Mouse circadian studies

Plasma, liver, epididymal adipose tissue and the MBH were collected from ad libitum-fed young mice every 4 h (starting at 10 a.m.) throughout two consecutive circadian cycles. We dissected the MBH in the anterior border of the optic chiasm, posterior border of the mammillary body, ventral third from the upper border of the anterior commissure, and medial third from the lateral border of the lateral sulcus in the ventral side of the brain. Leukocytes were isolated from pooled blood samples (three mice to one sample) using the Ficoll-Paque PLUS Cytiva (Sigma, 17-1440-02). In food-shift studies, food was available only during the light period (8 a.m. to 8 p.m.) for 14 days before sacrifice. In light-shift studies, mice were subjected to a reverse light/dark cycle (lights on from 8 p.m. to 8 a.m.) with free access to food for 2 weeks. Mice were sacrificed either at 2 a.m. or 2 p.m. to collect plasma and tissues. We repeated these experiments at least twice.

## Tissue NAD$^+$ measurements

Blocks of MBH, liver, and fat tissues were weighed, lysed with 150 µl of 1 M HClO$_4$, and neutralized by adding 50 µl of 3 M K$_2$CO$_2$, as previously described[14]. Small tissue blocks from 4 liver lobes were pooled to measure the liver NAD$^+$ levels. After centrifugation for 15 min (4 °C, 13,000 × g), 100 µl of supernatant was mixed with 30 µl of 50 mM K$_2$PO$_4$/50 mM KHPO$_4$ (1:1 v/v, pH 7.0) and loaded onto the column (AHima HPC 18AQ 5 µM, 15 × 4.6 cm). HPLC was performed at a flow rate of 1 ml/min. NAD$^+$ was eluted as a sharp peak at 10 min. NAD$^+$ was quantitated based on the peak area compared with a standard curve and normalized to tissue wet weight. The tissue NAD$^+$ levels were also measured using a NAD assay kit (Bioassay systems, E2ND-100). Homogenized tissue samples were lysed with NAD$^+$ extraction buffer included in the kit and then subjected to heat extraction at 60 °C for 5 min. After centrifugation of samples, 40 µl of supernatant was used for NAD$^+$ analysis.

## NAMPT ELISA and enzymatic activity assay

Plasma and serum were collected from mice and humans, respectively, and stored at −70 °C until use. Mouse blood eNAMPT levels were measured using an ELISA kit (Abnova, KA3659) according to the manufacturer's instructions. A volume of 10−20 µl of mouse plasma was diluted (1:5 or 1:10) in lysis buffer immediately before the assay. Human blood eNAMPT levels were measured using a sandwich ELISA kit (Adipogen, AG-45A-0008). A volume of 100 µl of human serum was assayed without dilution. The lower and upper limits of detection were 0.12 and 8 ng/ml, respectively. The intra- and inter-assay coefficients of variation were 5.6% and 5.9%, respectively. Measurement of NAMPT enzyme activity was performed using a NAMPT activity assay kit (Abcam, 221819) and commercial protocol. Enzyme activity was normalized by the protein amount. All samples were assayed in duplicates.

## Monitoring of metabolic behaviors

We measured circadian rhythms of LMA and EE using a Comprehensive Lab Animal Monitoring System (CLAMS, Columbus Instruments). This system measured oxygen consumption (VO$_2$) and carbon dioxide production (VCO$_2$) using an indirect calorimeter and we calculated EE by using the manufacturer's formula. Light and feeding conditions were identical to the home cage conditions unless indicated otherwise. Mice were adapted to the CLAMS cages for 48 h before studies.

## FK866 osmotic pump study

We infused FK866 to mice using an osmotic mini-pump (Alzet, 1003D) placed in the peritoneum. FK866 (3.3 and 8 mg/kg/day) was dissolved in a mixture of DMSO-saline (1:10 v/v) and infused at a flow rate of 0.5 µl/h for 3 days. Control animals received the same volume of vehicle. During infusion, mice were placed in CLAMS chambers to monitor LMA, EE, and FI. On the final day of the study, animals were sacrificed every 4 h throughout a circadian cycle to collect plasma, liver, MBH and white blood cells.

## Systemic Nampt siRNA study

We randomly assigned animals to receive either control or Nampt small inhibitory RNA (siRNA) one day before the study to match the average body weights between the two groups. Mice received siRNA targeting murine Nampt (33 nmol/kg, resuspended in 100 µl of RNase-free PBS; Dharmacon; E-040272-00-0005) via tail vein injections at -10 a.m. The control group received an equal amount of non-targeting, scrambled, control siRNA (Dharmacon; D-001910-10-20). After siRNA injections, we monitored LMA, EE, and FI until sacrifice at 2 a.m. (ZT18) on the 3rd day. We collected samples from white blood cells, plasma, hypothalamus, liver, and epididymal fat. Knockdown was considered successful if there was a >50% decrease in plasma eNAMPT concentrations compared with the average levels of controls at the end of the study. Only the animals demonstrating successful knockdown were included in the data analysis.

## NAMPT and NMN injection studies

We administered NAMPT protein and NMN at indicated doses i.p. (100 µl volume) or i.c.v. (2 µl volume) to C57BL/6J mice in the mid-light phase (ZT6−ZT8). Control mice received the same volume of vehicle. After a 48-h acclimatization period, we monitored LMA, EE, and FI for 4 h following injections using CLAMS. For the i.c.v. injection study, we implanted permanent 26-gauge stainless steel cannulae into the third ventricle (1.8 mm caudal to bregma, 5.0 mm ventral to the sagittal sinus) by using a stereotaxic surgery. One week after surgery, we confirmed positioning of the cannulae by a positive dipsogenic response to angiotensin II (50 ng, i.c.v.) and then daily handled mice for 1 week to minimize stress response. We used only the mice with correctly positioned cannulae for the study. We randomly assigned mice to the control or experimental groups 1 day before the study to match the average body weights between groups. We dissolved NAMPT, SHU9119, and MG132 in 0.9% saline prior to injection and administered at a volume of 2 µl.

## Organotypic slice culture and real-time bioluminescence imaging

Preparation of organotypic SCN cultures and bioluminescence monitoring were carried out as previously reported with minor modifications[60]. Neonatal (5−7 days old) PER2:LUC knock-in mice were sacrificed, and the brains were quickly obtained. The isolated brains were chilled in ice-cold Gey's balanced salt solution (GBSS) supplemented with 0.01 M HEPES and 36 mM D-glucose, bubbled with 5% CO$_2$ and 95% O$_2$, and then coronally sectioned into 400-µm-thick slices with a vibratome (Campden Instruments). The slices were maintained on a culture insert membrane (Millicell-CM, Millipore) and placed in a culture medium (50% minimum essential medium, 25% GBSS, 25% horse serum, 36 mM D-glucose, and 1x Antibiotic-Antimycotic solution) at 37 °C. The SCN explants were cultivated for more than 2 weeks before being used in experiments. All culture media and supplements

were purchased from Thermo-Fisher Scientific Co. unless specified otherwise. For bioluminescence recording, SCN cultures from the PER2:LUC mice were maintained in sealed 35-mm petri dishes with 1 ml of culture medium containing 0.3 mM D-luciferin (Promega) at 36 °C. Light emission was measured and integrated for 1 min at 10-min intervals with a dish-type wheeled luminometer (AB-2550 Kronos-Dio, ATTO). The background-subtracted bioluminescence profiles were analyzed using the Cosinor analysis software (http://www.circadian.org).

## c-Fos studies

POMC-tdTomato mice and NPY-hrGFP mice received injections of either NAMPT (1 mg/kg, i.p.) at 1 p.m. (ZT5) or FK866 (10 mg/kg, i.p.) at 1 a.m. (ZT17). One hour after injection, mice were transcardially perfused with 4% paraformaldehyde (PFA) for 15 min using a peristaltic pump. Whole brains were collected, post-fixed with 4% PFA overnight, and kept in 30% sucrose for 48 h. Brain sections (20 µm) were obtained and incubated overnight at 4 °C with a rabbit anti-c-Fos antibody (1:1000, Synaptic Systems, #226 003). Tissues were rinsed with PBS and then incubated with Alexa Fluor 488-conjugated anti-rabbit IgG (1:500, Invitrogen) at room temperature for 1 h. After washing with PBS, the slices were mounted using Vectashield mounting media (Vector Laboratories) and viewed using an LSM 700 confocal microscope (Carl Zeiss). Three matched brain sections, which included the ARC, were analyzed in each animal. The numbers of POMC and NPY/AgRP neurons with nuclear c-Fos immunoreactivity were counted and expressed as a percentage of the total number of POMC or NPY/AgRP neurons. c-Fos positive neurons were counted using ZEN microscope software (version 2.1 blue edition, Carl Zeiss).

## Hypothalamic FOXO1 adenovirus study

Adenoviruses expressing FOXO1-GFP fusing gene (Foxo1-Ad, $1 \times 10^{11}$ plaque-forming unit) were microinjected bilaterally into the mediobasal hypothalamus (5.7 mm depth, 1.8 mm caudal to the bregma, 0.3 mm lateral from the sagittal suture) via a syringe pump (Harvard Apparatus) at a rate of 100 nl/min for 5 min (0.5 µl/injection site)[42]. Animals were randomly assigned to experimental groups according to their body weight. Control animals received the same amount of GFP-Ad. On the third day of adenovirus injection, we monitored LMA, EE, and FI following i.c.v. administration of NAMPT or saline. Appropriate injection of the adenovirus was verified by the presence of green fluorescence in the MBH under a confocal fluorescence microscope at the end of the study. (Supplementary Fig. 18a). Results from animals that were shown to have received appropriate injections were included in the analyses. As a separate experiment, mice injected with GFP-Ad or Foxo1-Ad were sacrificed 2 h after i.c.v. injection of saline or NAMPT (100 ng) on the third day after adenovirus injection to determine the hypothalamic neuropeptide expression. In this experiment, we assessed successful viral injection with increased *Foxo1* expression (Supplementary Fig. 18b).

## Hypothalamic siRNA study

siRNAs specific to murine/human Sirt1 and Sirt2 were purchased from Dharmacon (M-061727-01 for mouse Sirt2; M-004826-02 for human SIRT2, E-003540-00 for human SIRT1). For the cell study, SH-SY-5Y cells were transfected with siControl, siSIRT1 or siSIRT2 (50 nM each). 48 h later, total RNA and protein were extracted to check for successful knockdown of the target gene (Supplementary Figs. 12a and 14b). For the animal study, murine siSirt2 was resuspended in RNase-free water and mixed with lipofectamine (9:1 v/v, Invitrogen) to a final concentration of 1 mM. Sirt2 siRNA was injected bilaterally (0.5 µl each side) into the ARC over a 5 min period as previously described[42]. The same amount of non-targeting scrambled control siRNA (Dharmacon; D-001910-10-20) was administered to the control group. On the second day of siRNA injection, NAMPT (100 ng) or saline was administered via the i.c.v.-implanted cannulae at ZT6 for the LMA, EE, and FI study. On the 3rd day, mice were sacrificed at ZT6 (1 h after i.c.v. saline or NAMPT injections) to collect the MBH. Animals showing a successful gene knockdown (below 50% of the average levels of controls) were included in the data analysis. The experiments were repeated at least twice.

## Measurement of mRNA expression

I.c.v. administration of NAMPT (100 ng) was carried out at ZT16. MBH tissue blocks were collected 2 h after NAMPT injection, and total RNA was extracted from the MBH using Trizol reagent (Invitrogen) and was quantified by spectrophotometry. To analyze hypothalamic region-specific gene expression, SCN, ARC and DMN tissues were micro-dissected from 10 µm-thick hypothalamic slices using a laser micro-dissection system (Leica LMD6500) and RNA was extracted using QIAGEN RNeasy Micro Kit (#74004). The mRNA levels of genes were determined using real-time PCR analysis (Perkin-Elmer) using the appropriate primers (Supplementary Table 5). The expression level of each mRNA was normalized to that of glyceraldehyde 3-phosphate dehydrogenase (Gapdh) or ß-actin mRNA.

## Immunoblotting and quantification

Mice were sacrificed by decapitation under the indicated conditions. Plasma and tissues were collected, immediately frozen in liquid nitrogen, and stored at −70 °C until use. Tissue lysates (50 µg protein) were separated by 8% SDS-PAGE and transferred to a PVDF membrane (GE Healthcare). Following incubation in blocking buffer, membranes were incubated overnight at 4 °C with antibodies against NAMPT (1:1500, mouse, Adipogen, AG-20A-0034), FOXO1 (1:1000, rabbit, Cell Signaling, 2880S), SIRT1 (1:1000, mouse, Santa Cruz, sc-74465), and SIRT2 (1:1000, rabbit, Cell Signaling, 12650). A 2-µl volume of plasma was diluted in 18 µl of lysis buffer and then subjected to immunoblotting for eNAMPT. Plasma protein loading was normalized using transferrin (1:1000, rabbit, Abcam, ab82411). For the immunoprecipitation assay, we extracted proteins from mouse hypothalamus or from cells transfected with Flag-tagged FOXO1- and SIRT2-expressing plasmids (1 µg each). We then incubated protein extracts (500 µg) with 1 µg of primary antibodies against FOXO1 (Cell Signaling, 2880 S), Flag (Sigma, F3165) or an equivalent amount of rabbit immunoglobulin-γ (IgG) overnight before adding 100 µl agarose bead (Roche). Subsequently, we performed immunoblotting using antibodies specific to SIRT2, FOXO1, and acetylated lysine (1:1000, rabbit, Cell Signaling, 9441). Blots were developed using a densitometer (VersaDoc Multi Imaging Analyzer System, Bio-Rad) and normalized using β-actin immunoblotting (1:1000, mouse, Santa Cruz, sc-47778). We quantified fluorescence intensity using Image J (version 1.52p, NIH).

## FISH and IF staining

Whole brains were collected from cardiac-perfused mice with 50 ml saline followed by 50 ml 4% PFA in DEPC water via the left ventricle of the heart under anesthesia induced with 5 mg/kg Rompun and 40 mg/kg Zoletil. Brains were post-fixed with 4% PFA at 4 °C overnight, and kept in 30% sucrose solution until the tissues sank to the bottom of the container. Brain sections (14-µm thick) including the hypothalamus were generated using a cryostat (Leica). Brain slices were incubated with antigen retrieval and protease agent, and hybridized with Sirt2 probes (ACDBio, 1161441-C1) for 2 h at 40 °C, followed by step-wise amplification according to the manufacturer protocol. Following the completion of RNA in situ hybridization, brain sections were subjected to immunofluorescence staining by incubation with 3% BSA in PBS for 1 h and then with diluted POMC (ß-endorphin) antibody (1:1000, rabbit, Phoenix, H02233) in 3% BSA in PBS overnight at 4 °C. After washing, slides were incubated with the appropriate Alexa-Flour 488-conjugated secondary antibodies (1:1000) at room temperature for 2 h. Immunofluorescence images

were obtained using a confocal microscope (Carl Zeiss 710, Germany). For NPY and *Sirt2* double staining, brains were obtained from NPY-hrGFP mouse and subjected to *Sirt2* in situ hybridization and followed by immunostaining using GFP antibody (1:1000, goat, Abcam, ab6673) as described above.

## Promoter assay

The promoter regions of the human *AGRP* gene (nucleotide positions −1000 to −1) or *POMC* gene (nucleotide positions −877 to +320) were cloned and ligated into a luciferase reporter as previously described[42]. SH-SY5Y cells were cultured in 24-well plates and were transfected with *pAGRP-luc* or *pPOMC-luc* reporters (50 ng) with or without human *SIRT2* (100 ng), human wild-type *FOXO1* and *FOXO1-6KR* (100 ng each, gift from Dr. Tadahiro Kitamura at Gumma University), *siControl*, *siSIRT1*, *siSIRT2* (50 nM each), or *CMV-β-gal* (25 ng) using lipofectamine. At 48 h after transfection, SH-SY5Y cells were treated with NAMPT protein (1, 10, and 100 nM) for 2 h following 2 h-serum starvation. At the end of the study, cells were harvested to measure luciferase activities, which were normalized to those of β-galactosidase. Transfections in each condition were performed in at least triplicates and the experiments were repeated three times. Data are shown as fold changes compared with mock vector-transfected controls.

## Human eNAMPT study

For human studies, 14 healthy Koreans (10 males, 4 females) aged 18–30 years were recruited via local website advertising. Seven subjects (5 males, 2 females) were lean (BMI < 23 kg/m$^2$) and 7 subjects (5 males, 2 females) were individuals with obesity (BMI ≥ 25 kg/m$^2$), according to the definition of obesity from the International Diabetes Federation−Western Pacific Region. All subjects signed an informed consent form. This study was approved by the Institutional Review Board of Asan Medical Center (Seoul, Korea) (IRB approval number: 2008-0314, approved date: August 11, 2008) and carried out in accordance with the principles of the Declaration of Helsinki. Subjects were advised to sleep between 10 p.m. and 6 a.m. for 1 week before the study. All subjects were admitted to the Asan Clinical Research Center at 5 p.m. on the day before the study day and were provided with dinner at 6 p.m. An in-dwelling intravenous catheter was inserted into the forearm vein for blood sampling after dinner. On the study day, blood was drawn into serum-separating tubes containing aprotinin (250 kallikrein inhibitor units; Bayer) at 30-min intervals from 6 a.m. to 10 p.m. Thereafter, samples were drawn hourly until 6 a.m. on the next morning. Subjects had breakfast at 8 a.m., lunch at 12:30 p.m., and dinner at 6 p.m., which consisted of 60% carbohydrates, 25% fat, and 15% protein. Lights were off from 10 p.m. to 6 a.m.; the lights were on at all other hours. Subjects were allowed a rest period except when eating and using the lavatory.

## Blinded research

Investigators were not blinded in conducting animal experiments or collecting samples for analysis. Where possible, researchers were blinded during data analysis.

## Statistical analysis

Statistical analysis was performed using SPSS version 24 (IBM Analytics) and Graphpad Prism version 7. Statistical significance between groups was determined using one- or two-way analysis of variance, followed by a post hoc Fisher's least significant difference test and an unpaired Student's *t*-test, as appropriate. JTK-CYCLE algorithm ('MetaCycle' package of R software, version 4.1.1) was used to detect circadian rhythmicity and circadian rhythm parameters were assessed using the cosinor package of R software (version 4.1.3; R Foundation for Statistical Computing, Vienna, Austria). Wald test was used to compare amplitude and acrophase between lean and obese mice while Student's *t*-test was used for the comparison of mesor. Statistical

significance was defined at $P < 0.05$. Data are presented as mean ± standard error of the mean (SEM).

## Reporting summary

Further information on research design is available in the Nature Portfolio Reporting Summary linked to this article.

## Data availability

All data generated or analyzed during this study are provided with this paper in the source data files. Source data are provided with this paper.

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

## Acknowledgements

This study was supported by grants from the National Research Foundation of Korea (NRF) funded by the Ministry of Science and ICT of Korea (2020R1A2C3004843, 2020R1A4A3078962, 2022R1C1C1012590, 2022R1C1C2007378, and 2022M3E5E8017213). We appreciate Dr. Min-Ju Kim at the Department of Clinical Epidemiology and Biostatistics, Asan Medical Center for statistical analysis of circadian rhythms.

## Author contributions

J.W.P., S.Y.G., E.R., G.H.S., and M.S.K. were involved in the experimental design; J.W.P., E.R., G.M.K., S.Y.G., H.K.K., C.H.L., W.H.J., S.E.P., S.Y.M., S.K., S.Y.J., C.B.P., H.S.L., Y.R.O., and H.N.J. performed the experiments; O.K., B.S.Y., G.H.S., and S.H.M., discussed the data; J.W.P. S.H.M., and M.S.K. analyzed the data and wrote the manuscript; M.S.K. directed the study.

## Competing interests

The authors declare no competing interests.
