## [Peer Review File · Nature Communications]

Reviewers' Comments:

Reviewer #1:

Remarks to the Author:

NAD⁺ is an essential compound used in many cellular redox reactions to maintain a variety of metabolic functions, such as fatty acid beta oxidation, the tricarboxylic acid cycle, and glycolysis. In mammals, intracellular NAD⁺ biosynthesis in most tissues is controlled by the salvage pathway dependently of the rate-limiting enzyme NAMPT, specifically the extracellular NAMPT (eNAMPT) secreted from peripheral tissues. SIRT1 is an NAD⁺-dependent sirtuin family deacetylase, which directly links cyclic biological rhythms and energy metabolism in the cell. Recently, a large body of studies have showed that NAD⁺ oscillation plays an important role in driving hypothalamic neuronal functions. In this study, the authors have investigated the hypothalamic eNAMPT-NAD⁺-SIRT axis in the circadian regulation of energy metabolism. The author reported that, via the NAD⁺-SIRT2-FOXO1 pathway, hypothalamic NAD⁺ shows the circadian oscillation in phase with plasma eNAMPT; changes of eNAMPT levels are capable of altering circadian hypothalamic NAD⁺, locomotor activity and energy expenditure (EE); and obese humans and animals blunts those changes. Despite the work carried out by the authors is certainly of interest (especially, human studies were also included), it also generates a number of comments, which are further detailed below.

Main comments:

1. In Fig.1, the authors found that early morning (ZT2) peak of hypothalamic NAD⁺ is in accord with that of hypothalamic iNAMPT and the major night peak (ZT18) of hypothalamic NAD⁺ is in phase with that of plasma eNAMPT. Thus, their claim of "the diurnal rhythm of hypothalamic NAD⁺ levels is consistent with that of plasma eNAMPT" is not accurate. Also, the authors used the MBH blocks as their study tissues, which (the authors didn't mention how they collected) may contain the Arc, DMH, VMH, and probably SCN parts. As SIRT functions in different region of the hypothalamus are physiological different or contrast (i.e. VMH vs Arc), which may cause 'interference' to their results, and their following and main findings were focused on Arc area (i.e. no changes were found in DMH in Fig.4B), I just wondering why the authors didn't collect the Arc part as their study target with the more precise approach, such as laser microdissection?
2. The authors reported that i.p. infusion of FK866, the NAMPT inhibitor, increases the nocturnal food intake in tested mice (Suppl. Fig.2). This result is opposite to the previous finding (i.e. *Acta Physiol (Oxf)*. 2020; 228(4): e13437) that reported FK866 administration diminishes food intake during the dark phase and ghrelin-induced hyperphagia. Also, the authors didn't report/find this hyperphagic effect following reduction of eNAMPT with siNampt approach, but reported a suppression on fasting-refeeding upon systemic or central elevation of NAMPT.
3. The analyses of locomotion and EE in Fig. 2 are different from other figures (AUC vs. others?).
4. The representative Arc images presented for demonstrating c-fos expressions in Fig.4 are not matched (not at the same rostro-caudal level). And, the unbiased quantify information of c-fos counting was not clearly provided.
5. SIRT2, instead of SIRT1, has been shown in this manuscript as the downstream mediator of eNAMPT. Although the co-staining results showed that SIRT2 co-expresses with POMC and NPY/AgRP and SIRT2 deletion blunted NAMPT's actions, the authors didn't show whether SIRT2 can directly regulate POMC or NPY/AgRP?
6. In the FOXO1 overexpression experiment, the authors described that their stereotaxic surgeries for Arc injection are at 2.8mm caudal to the bregma. According to the mouse atlas and our experience, this coordinate is far too caudal from where the hypothalamus is located. Also, the figures (suppl. Fig.8) showed of the FOXO1 expression are at around AP -2.30mm level. The authors need to carefully check this part.
7. In Fig.6i, the group of FOXO1 overexpression only should be included.
8. The main behavioral parameters measured in this study were locomotion, EE, and food intake by using the CLAMS system. Given the short/no window adapted by the authors following experimental treatments (i.e. surgeries for pump delivery of FK866, or Arc stereotaxic injections, etc.), I just wondering how the authors rule out the significant impact of these big 'stressors' on testing animals' behaviors, such as feeding and locomotion, in such a short time?
9. The description of experimental design, flow, and time line are very confusing and hard to be followed, the authors need to extensively improve it.

Minor comments:

1. The Zeitgeber time (ZT) and actual time were both used to present data (i.e. Fig.1g and h, and suppl. Fig.1) in this study, which is very confusing. The authors should use only the ZT for labelling all the figures.
2. In Fig.2 b and c, the pink and red colors are not distinctive enough for easy read. It would be better if the authors can use the clearly contrastive colors.
3. ARGP should be replaced with AgRP in the manuscript.

Reviewer #2:

Remarks to the Author:

In this study, the authors found that the circadian eNAMPT levels in blood were in phase with hypothalamic circadian NAD⁺ contents in mice. Furthermore, hypothalamic circadian NAD⁺ levels produced by eNAMPT activity regulate locomotor activity and energy expenditure via the hypothalamic NAD⁺/SIRT2/FOXO1/melanocortin axis. Overall, the data are clear and solid, but there is some critical data/information required, which will improve this paper.

Major comments

1. The authors use "diurnal" in the text as the meaning of "a daily". However, in the circadian research fields, "diurnal" is often used as "a peak during the daytime" and "nocturnal" is used as "a peak during the nighttime". Actually, nocturnal is used in this text. That might make the readers and this reviewer confuse. This reviewer suggests using daily, circadian, or a time-of-the-day, instead of "diurnal".
2. Line148-; This reviewer recommends the authors investigate whether and how Nampt mRNA expressions oscillate in leukocytes. This result might clarify the involvement of eNAMPT from leukocytes for hypothalamic circadian NAD⁺ levels.
3. Line 155; The authors should measure exogenous plasma NAMPT concentration to emphasize that ip-injected NAMPT (0.3 mg/kg) makes plasma NAMPT concentration much more than endogenous eNAMPT. The authors describe in the text that eNAMPT possesses higher activity than iNAMPT (line 49), and use recombinant NAMPT, which activity considered to be similar to iNAMPT, for injection.
4. Fig.5c and d; Show the data on whether the 2hr-NAMPT treatment is enough to increase cellular NAD⁺ levels. eNAMPT is also known as visfatin, and that can stimulate MAPK and PI3K/AKT pathways, which activations may regulate POMC and/or AGRP gene expressions. This reviewer considers that NMN treatment is enough to investigate this, instead of the NAMPT treatment.
5. The authors should statistically analyze data by JTK_cycle or related ones, not ANOVA whether daily NAD⁺ or NAMPT levels show circadian patterns. Check this article; Hughes et al., J Biol Rhythms. 2010 Oct;25(5):372-80. doi: 10.1177/0748730410379711.

Minor comments

1. Line 56; Should be "forkhead".
2. Line 84; iNAMPT activity => levels, because the cited article #14 has compared the iNAMPT levels among tissues, not its activities.
3. Which experiments are AtT20 cells used for? There are no descriptions. Also, cell line information is missing in the figure legends.
4. Line 13; Full-length NAMPT is not a peptide, "protein".
5. Line 160-161; This reviewer does not understand the sentence, "The negligible effect ...". Because NAD⁺ amounts in MBH and liver under control conditions are almost the same, 180 pmol/mg tissue, and eNAMPT and iNAMPT function completely in different regions. That suggests i.p. NAMPT should also affect NAD⁺ amount even in the liver. Explain this.
6. Line 237; Based on the authors' experiment, they cannot say "direct" binding.
7. Line 238; "acetylated form of FOXO1" is better instead of "expression of acetylated FOXO1".
8. No description of Fig.7b.

9. Fig.1d; Compared to iNAMPT, for this reviewer eNAMPT seems to possess circadian post-translational modifications. This reviewer recommends the authors discuss this.
10. Fig.2 b and c; black and white bars are opposite.
11. Fig.2e and h; iNAMPT amounts in the liver are decreased by siNampt treatment, but why are liver NAD⁺ amounts not changed? Discuss this.
12. Fig.2 c and g; In controls, Fig.2c shows high EE during the daytime, but Fig.2g shows the opposite result.
13. Fig.5c and d; To avoid the possibility that SIRT1 is associated with POMC/AGRP gene expressions, it would be better to knock down Sirt1 and check these gene regulations.
14. Fig.7a; MBH and plasma NAD⁺ amounts in DIO mice seem to oscillate for this reviewer. Perform statistical analyses.
15. No description of per2::luc mice in "Animals"/METHODS.

Reviewer #3:

Remarks to the Author:

This is an interesting paper in which the authors explore the role of eNAMPT in the hypothalamic control of energy balance. Using behavioral, physiological and molecular techniques, the authors found that the eNAMPT-NAD⁺ pathway is very important in the hypothalamic control of energy balance. Although similar results had been shown previously (Yoon et. Al. (2015), in this work the authors propose a specific mechanism in POMC and AGRP neurons, on this control. The authors demonstrate that the NAD⁺-dependent activity of SIRT2 is of central importance in controlling the activity of the transcription factor FOXO1, which in turn would control the expression of POMC and AGRP by inhibiting and inducing their expression respectively. The authors further explored NAD⁺ levels in DIO mice, arguing that in this condition NAD⁺ oscillation lost robustness. Finally, they explored eNAMPT levels in the blood of lean and obese subjects. Although these results are not conclusive, lean subjects appear to have better oscillations in eNAMPT. Although the paper is extensive and used a wide variety of techniques some results seem to be not very conclusive as they are, therefore, I suggest reorganizing the figures and results and adding some analysis including feeding and analytical reinterpretations and also improving the discussion section

Please consider the following observations.

LINE 75 as for definition zt0=lights (at 8am) this info can be included in methods

LINE 90 "These findings suggest that a rise in hypothalamic NAD⁺ contents during the dark period may be driven by plasma eNAMPT, whereas its early-morning rise may be driven by MBH iNAMPT." I's not clear why the authors suggest this.

LINE 97 "In the epididymal adipose tissue, diurnal fluctuations of iNAMPT expression were less distinct than those in the liver and plasma of freely-fed male mice"
this is not clear, similar to blood and liver? It's seem the opposite, a reduction during the dark phase

LINE 124 "EE, and FI. eNAMPT inhibition was induced by continuous infusion of FK866"
how to know if the iNAMPT was not inhibited as well?

LINE 125 "In mice receiving FK866 infusion (8 mg/kg), nocturnal rises in the MBH NAD⁺ levels"
how did the authors calculate this concentration?

LINE 126 "were abolished and the normal circadian patterns"
with only two ZT points it is difficult to assume a circadian pattern

LINE 128 "These data suggest that normal diurnal variations in hypothalamic NAD⁺"
this is by humoral NAD⁺ levels?

LINE1 131 "markedly blunted the nighttime increases in LMA and EE"
the reduced effect in EE is by the reduce LMA? Please discuss this.

LINE 132 "during the dark" dark period

LINE 136 "siNampt (33 nmol/kg) in the tail vein reduced plasma eNAMPT by ~40% and MBH

NAD⁺ levels"

what tissue could be affected in the salvage pathway? WB experiments in the liver, hypothalamus and adipose tissue would be required

LINE 142 "siNampt-induced depletion of blood eNAMPT levels recapitulates the effects of chemical NAMPT inhibition on LMA and EE"

what is known about the effects of NAD⁺ on activity, feeding and EE? because feeding is rather opposed to EE and activity (excepting the activity induced prior to feeding in a RF paradigm known as Feeding anticipatory activity)

LINE 147 "dark" dark period

LINE 153 "daytime" rather light phase

LINE 155 "endogenous plasma eNAMPT activity" activity? or levels? the authors should provide a reference/figure

LINE 167 "On the other hand, intraperitoneally injected NAMPT/NMN and ICV-administered NAMPT suppressed fasting-induced hyperphagia"

it should be discussed the role of NAD/NMN on energy balance

LINE 168 "These findings support that circulating eNAMPT is critical for generating the day-night rhythms in LMA and EE." this is not clear because there are not experiments on the dark phase

LINE 193 "The percentages of c-Fos+ POMC neurons were significantly different between ZT6 and ZT18 (13% vs. 28%, P = 0.04)"

POMC is an anorexigenic peptide how to explain the increased activity of these neurons at ZT 18 vs ZT6

LINE 194 "variation" add "in the activity"

LINE 195 "while the high POMC neuronal activity at ZT18 was suppressed by FK866"

how relates this with the natural rhymes in the hypothalamic NAD⁺?

LINE 203 "POMC end product α -melanocyte stimulating hormone (α MSH) and AGRP regulate LMA and EE, and these effects may be partly via their agonistic and antagonistic actions on melanocortin 3 and 4 receptors (MC3/4R)³⁴"

more references are needed

LINE 213 "the circadian oscillations of hypothalamic SIRT1 expression" the circadian expression of Sirt1, has not been clearly demonstrated, rather its activity induced by the oscillating NAD⁺ provokes circadian oscillation in its deacetylase activity and therefore on the acetylation levels on its target's proteins. Therefore, the result should be taken with cation. The authors might discuss this.

LINE 214 "marked diurnal rhythm in SIRT1 expression that peaked at ZT6–10 and decreased at ZT18"

To get a more precise analysis, a cosinor analysis should be done, to obtain the acrophase and amplitude parameters.

LINE 219 "Sirt2/NPY-GFP confirmed Sirt2 expression in both types of neurons (Fig. 5b), and in vitro promoter assay revealed that NAMPT treatment increased the POMC transcriptional activity but repressed the AGRP transcriptional activity"

since SIRT2 is a deacetylase enzyme, what could be the epigenetic mechanism?

LINE 227 "These findings suggested that hypothalamic SIRT2 acts downstream of eNAMPT in the diurnal modulation of LMA and EE involving POMC and NPY/AGRP neurons"

1) what is the effect of NPY or POMC on the circadian activity?, 2) the oscillation of POMC or NPY/AGRP should be shown, 3) food intake is an very important parameter it should be shown, 3) It is not clear the inter or extra cytosolic NAD⁺ mechanism such as the extra to intracellular transportation, etc., need to be discussed.

LINE 238 "decreases the expression" rather the acetylation level

LINE 244 "Overexpression of FOXO1 deacetylation mutant (6KR)42"
what is the functional importance of this region/residue for the FOXO1 activity?

LINE 244 "deacetylation mutant" rather with a mutation on the amino acid that is deacetylated by SIRT2 (ref)

LINE 245 "effects of WT FOXO1 on AGRP and POMC transcriptional activity (Fig. 6d)."
This is confused, because it is not possible to differentiate between cellular and recombinant FOXO1 proteins, and how unacetylated FOXO1 decreases the level of FOXO1?, also how to know that the acetylation levels are different? it would be needed an antibody against acetylated residues.

LINE 254 "hypothalamic FOXO1 levels (Supplementary Fig. 7), suggesting that eNAMPT can promote hypothalamic FOXO1 degradation in vivo."
to include a reference on these mechanisms mediated by NAMPT/NADH or acetylation

LINE 257 "decrease in hypothalamic FOXO1 levels" at what ZT?

LINE 271 "blunted diurnal oscillation in MBH NAD⁺ contents, plasma eNAMPT, and liver iNAMPT expression levels (Fig. 7a)." The NAD⁺ levels from control diet feed mice should be included in the same plot also to include a detailed circadian analysis such as the cosinor analysis.

LINE 278 "The demographic characteristics of the subjects"
These are not demographic characteristics

LINE 280 "Blood samples were collected every 30 min from 7 a.m. to 9 p.m., and then hourly until 7 a.m. in the next morning"
There was a period of adaptation on this schedule?

LINE 287 "showed altered circadian patterns in serum eNAMPT levels with oscillations during the light-off period (Fig. 7d)"
It could be that the effect on the diurnal peaks compared with the night period is rather an effect on the sampling (day vs night).

LINE 292 "conditions, which might underlie the disruptions in the diurnal rhythms of physical activity or sleep/wake cycle in obese individuals"
The results as these are, do not support this conclusion additional data analysis should be done such as a cosinor analysis to extract the amplitude, acrophase and mesor parameters, and then to statistically compare between groups.

LINE 295 "This rhythm persists under continuous darkness and is profoundly disturbed by the reversal of the light-dark phase and daytime-restricted feeding"
This is weird, because the authors restricted the feeding during the day being in fact the sleeping timing in mice, therefore I would rather expect a blunted effect when the feeding is restricted during the dark period, therefore I suggest restricting the food when the mice are active.

LINE 299 "Supporting this possibility, eNAMPT has a high enzymatic NAMPT activity¹⁴ 299 and its blood levels rise during the night"
Please discuss what may be the source of the hypothalamic eNAMPT.

LINE 302 "These findings strongly suggest that eNAMPT-mediated regulation of hypothalamic NAD⁺ biosynthesis occurs in a circadian manner"
To be a strong suggestion the data should be taken at more ZT points, so I recommend eliminating "strong"

LINE 304 "diurnal fluctuation in 6-months-old mice although the degree of fluctuation was less than that in our study."
Do the authors mean amplitude for the degree of fluctuation?

LINE 305 "old mice" when compared...

LINE 308 "Consistent with the findings from rodents, humans also display a distinct diurnal rhythm in blood eNAMPT"

What means for distinct, compared to what?

LINE 310 "Frequent blood sampling in our human study involving young lean subjects also revealed pulsatile oscillations during the day"

These could be ultradian rhythms and not circadian rhythms, this should be discussed and/or reanalyzed

LINE 311 this looks like a result description rather than a discussion

LINE 303 "There is compelling evidence that the eNAMPT upregulates hypothalamic NAD+ 312 levels, but the detailed mechanisms remain elusive" Please include references and the participation of eNAMPT on the NAD biosynthesis should be discussed in more detail.

LINE 339 "Future studies are needed to examine whether hepatocytes secrete eNAMPT through sirtuin- and EV-dependent mechanisms as shown in adipocytes^{13,15}"

This data seems to contradict the contribution of adipose tissue on the hypothalamic levels of NAMPT shown by Yoon et. Al. (2015). How to explain this discrepancy?

LINE 348 "Furthermore, IP administration of NAMPT peptide and NMN, which imitated the condition of increased systemic eNAMPT activity "

At which time was injected the NMN and NAMPT peptide? does this peptide has enzymatic activity?

LINE 350 "In contrast, eNAMPT appears to have a dispensible role in regulating the physiological fasting/feeding rhythms"

Also, here this data looks contradictory with the data reported by Yoon et. Al. showing that eNAMPT its important to maintain hypothalamic NAD+ levels during fasting

LINE 352 "Instead, eNAMPT may be involved in satiety formation in the middle of the dark phase, considering that systemic eNAMT inhibition with FK866 increased food intake during the night"

What can be discussed about gender? Also, as already suggested a food intake experiment would be very relevant; such as circadian or at least 2-4 ZT food consumption and/or the measurement of food intake during fasting at night and at day periods

LINE 376 "These findings support the hypothesis that eNAMPT stimulates LMA and EE through SIRT2-mediated FOXO1 inhibition and the resulting functional changes in POMC and NPY/AGRP neurons".

It could be very nice if the authors discuss in more detail the mechanism of FOXO1-mediated regulation of POMC and NPY neurons

LINE 381 "In addition, we found that DIO mice had blunted circadian oscillations in the liver iNAMPT and plasma eNAMPT expression and MBH NAD+ levels"

This is not clear, I suggest to directly compare these oscillation under control and HFD and/or analyze the circadian parameters

LINE 382 "Moreover, DIO disrupted diurnal rhythms in hypothalamic FOXO1 expression and LMA" As in the last suggestion, to analyze the circadian parameters on the levels of FOXO1

FIGURE 1B To separate light/dark areas as done in the panel A

FIGURE 2A In liver NAD+ are increased at the night onset, how to explain the increase in the NAD+ levels when eNAMPT is inhibited?

FIGURE 2H Are there any WB for these results?

FIGURE 3A What about the dark period?

FIGURE 4A Mesor is not the amplitude,

FIGURE 4A These plots are not clear, blue=vehicle and red=NAMPT, but in the histograms, they

are grouped in pairs

FIGURE 6B What means -OX?

FIGURE 6B SIRT1 inhibits Fox1-mediated repression / expression of POMC / AGRP respectively does this match with the NAD⁺ levels maybe to show a plot

FIGURE 6B To include the molecular weights

FIGURE 6D In this experiment SIRT1 over-expression would be included

Responses to Reviewers' comments:

We deeply thank the reviewers for their helpful comments. We believe that our manuscript has been improved by their advice.

Reviewer #1 (Remarks to the Author):

NAD⁺ is an essential compound used in many cellular redox reactions to maintain a variety of metabolic functions, such as fatty acid beta oxidation, the tricarboxylic acid cycle, and glycolysis. In mammals, intracellular NAD⁺ biosynthesis in most tissues is controlled by the salvage pathway dependently of the rate-limiting enzyme NAMPT, specifically the extracellular NAMPT (eNAMPT) secreted from peripheral tissues. SIRT1 is an NAD⁺-dependent sirtuin family deacetylase, which directly links cyclic biological rhythms and energy metabolism in the cell. Recently, a large body of studies have showed that NAD⁺ oscillation plays an important role in driving hypothalamic neuronal functions. In this study, the authors have investigated the hypothalamic eNAMPT-NAD⁺-SIRT axis in the circadian regulation of energy metabolism. The author reported that, via the NAD⁺-SIRT2-FOXO1 pathway, hypothalamic NAD⁺ shows the circadian oscillation in phase with plasma eNAMPT; changes of eNAMPT levels are capable of altering circadian hypothalamic NAD⁺, locomotor activity and energy expenditure (EE); and obese humans and animals blunts those changes. Despite the work carried out by the authors is certainly of interest (especially, human studies were also included), it also generates a number of comments, which are further detailed below.

Main comments:

1. In Fig.1, the authors found that early morning (ZT2) peak of hypothalamic NAD⁺ is in accord with that of hypothalamic iNAMPT and the major night peak (ZT18) of hypothalamic NAD⁺ is in phase with that of plasma eNAMPT. Thus, their claim of “the diurnal rhythm of hypothalamic NAD⁺ levels is consistent with that of plasma eNAMPT” is not accurate.

Response: Thank you for reviewing our study in detail and providing critically helpful comments. According to the reviewer’s comment, we revised the title of Fig. 1 in the manuscript (Line 73) and Fig. 1 legend.

Also, the authors used the MBH blocks as their study tissues, which (they collected) may contain the Arc, DMH, VMH, and probably SCN parts.

Response: We described how we collected the MBH in the Method section (Line 502–506).

As SIRT functions in different region of the hypothalamus are physiological different or contrast (i.e. VMH vs Arc), which may cause ‘interference’ to their results, and their following and main findings were focused on Arc area (i.e. no changes were found in DMH in Fig.4B), I just wondering why the authors didn’t collect the Arc part as their study target with the more precise approach, such as laser microdissection?

Response: The reviewer raised an important point. Unfortunately, the amount of tissues acquired using laser microdissection or punch biopsy would have been too little to perform multiple types of assay such as NAD⁺ measurement, western blotting, and qPCR.

2. The authors reported that i.p. infusion of FK866, the NAMPT inhibitor, increases the nocturnal food intake in tested mice (Suppl. Fig.2). This result is opposite to the previous finding (i.e. Acta Physiol (Oxf). 2020; 228(4): e13437) that reported FK866 administration diminishes food intake during the dark phase and ghrelin-induced hyperphagia. Also, the authors didn’t report/find this hyperphagic effect

following reduction of eNAMPT with siNampt approach, but reported a suppression on fasting-refeeding upon systemic or central elevation of NAMPT.

Response: Thank you very much for the thoughtful comment. To address the reviewer's question, we conducted a feeding study in which we injected several doses of FK866 (i.e., 100, 400, and 1000 ng) before light-off and measured the food intake over a 24-h period. This experimental design is similar to that used in a previous paper (*Acta Physiol (Oxf)*. 2020; 228(4): e13437). In that paper, the authors administered FK866 at 1 nmol (~430 ng) and 3 nmol (~1300 ng). In our additional feeding experiment, ICV FK866 (100 and 1000 ng) significantly increased the nighttime food intake as shown below. Interestingly, the lowest dose of FK866 (100 ng) was most effective, especially at the delayed time points. Thus, in our setting, FK866 administration increased food intake. We are unsure as to what led to this discrepancy between our and the previous data. We have also presented the hyperphagic effect of Nampt siRNA treatment in Supplementary Fig. 3b.

3. The analyses of locomotion and EE in Fig. 2 are different from other figures (AUC vs. others?).

Response: We presented the data of LMA and EE over circadian cycles and AUC (area under the curve) in the light and dark period in Fig. 2. The effects of NAMPT treatment on LMA and EE lasted for less than 4 hours. Thus, we showed the data of LMA and EE as values per hour in other figures.

4. The representative Arc images presented for demonstrating c-fos expressions in Fig.4 are not matched (not at the same rostro-caudal level). And, the unbiased quantify information of c-fos counting was not clearly provided.

Response: As per your comment, we presented the rostro-caudal matched images in revised Fig. 4d, e. We counted the c-fos⁺ cell numbers in 3 rostro-caudal matched brain slices including the ARC per mice. We described how we counted c-fos⁺ cells in the Methods section (Line 600–601).

5. SIRT2, instead of SIRT1, has been shown in this manuscript as the downstream mediator of eNAMPT. Although the co-staining results showed that SIRT2 co-expresses with POMC and NPY/AgRP and SIRT2 deletion blunted NAMPT's actions, the authors didn't show whether SIRT2 can directly regulate POMC or NPY/AgRP?

Response: To address the reviewer's comment, we conducted POMC and AgRP promoter assay in SIRT2 alone-overexpressing cells and showed the data in revised Fig. 6c (the third group). We found that SIRT2 alone-overexpression mildly but significantly increased POMC transcription but suppressed AgRP transcription. We described these findings in Line 277–278.

6. In the FOXO1 overexpression experiment, the authors described that their stereotaxic surgeries for Arc injection are at 2.8 mm caudal to the bregma. According to the mouse atlas and our experience, this coordinate is far too caudal from where the hypothalamus is located. Also, the figures (suppl. Fig.8)

showed of the FOXO1 expression are at around AP -2.30 mm level. The authors need to carefully check this part.

Response: We appreciate the reviewer for pointing out our mistake. We injected the virus targeting 1.8 mm caudal to the bregma and revised the sentence as such (Line 607).

7. In Fig.6i, the group of FOXO1 overexpression only should be included.

Response: We actually had the group of FOXO1 overexpression only but did not show the data. We now presented this data in Supplementary Fig. 17 and described the results in the text (Line 310–311).

8. The main behavioral parameters measured in this study were locomotion, EE, and food intake by using the CLAMS system. Given the short/no window adapted by the authors following experimental treatments (i.e. surgeries for pump delivery of FK866, or Arc stereotaxic injections, etc.), I just wondering how the authors rule out the significant impact of these big ‘stressors’ on testing animals’ behaviors, such as feeding and locomotion, in such a short time?

Response: As you commented, the effect of stress on animal’s behaviors is always of concern. The best we can do is to conduct experiments by skilled and experienced researchers and to compare them with controls exposed to the same stress. In our study, we adjusted the animals to the CLAMS chamber for at least 48 hours before the studies. In data analysis, we did not include the data obtained during the first 24 hours of post-surgical period and the data of sick animals with excessive weight loss.

9. The description of experimental design, flow, and time line are very confusing and hard to be followed, the authors need to extensively improve it.

Response: To help the reader’s understanding, we drew a schematic diagram of experimental design combined with the results in the figures.

Minor comments:

1. The Zeitgeber time (ZT) and actual time were both used to present data (i.e. Fig.1g and h, and suppl. Fig.1) in this study, which is very confusing. The authors should use only the ZT for labelling all the figures.

Response: Following the reviewer’ advice, we presented the time in ZT in all figures except Fig. 1b, which was conducted under continuous darkness.

2. In Fig.2 b and c, the pink and red colors are not distinctive enough for easy read. It would be better if the authors can use the clearly contrastive colors.

Response: We changed the bar colors in revised Fig. 2b, c.

3. ARGP should be replaced with AgRP in the manuscript.

Response: We replaced AGRP with AgRP. Thank you for the keen comment.

Thank you again for reviewing our study in detail and providing helpful comments. We hope that our responses and the corresponding revisions are satisfactory.

Reviewer #2 (Remarks to the Author):

In this study, the authors found that the circadian eNAMPT levels in blood were in phase with hypothalamic circadian NAD⁺ contents in mice. Furthermore, hypothalamic circadian NAD⁺ levels produced by eNAMPT activity regulate locomotor activity and energy expenditure via the hypothalamic NAD⁺/SIRT2/FOXO1/melanocortin axis. Overall, the data are clear and solid, but there is some critical data/information required, which will improve this paper.

Major comments

1. The authors use “diurnal” in the text as the meaning of “a daily”. However, in the circadian research fields, “diurnal” is often used as “a peak during the daytime” and “nocturnal” is used as “a peak during the nighttime”. Actually, nocturnal is used in this text. That might make the readers and this reviewer confuse. This reviewer suggests using daily, circadian, or a time-of-the-day, instead of “diurnal”.

Response: Thank you for reviewing our study in detail and providing critically helpful comments. We used “diurnal” instead of “circadian”. To avoid confusion, we changed “diurnal” to “circadian” throughout the text.

2. Line148-; This reviewer recommends the authors investigate whether and how Nampt mRNA expressions oscillate in leukocytes. This result might clarify the involvement of eNAMPT from leukocytes for hypothalamic circadian NAD⁺ levels.

Response: We measured the leukocyte Nampt mRNA expression levels over the circadian cycle and presented the data in Fig. 1g. Notably, leukocyte Nampt mRNA expression showed a distinct circadian rhythm with a peak at ZT10.

3. Line 155; The authors should measure exogenous plasma NAMPT concentration to emphasize that ip-injected NAMPT (0.3 mg/kg) makes plasma NAMPT concentration much more than endogenous eNAMPT. The authors describe in the text that eNAMPT possesses higher activity than iNAMPT (line 49), and use recombinant NAMPT, which activity considered to be similar to iNAMPT, for injection.

Response: We measured the enzyme activity of NAMPT protein with a commercial NAMPT activity assay kit. As shown in Supplementary Fig 5, this protein had a good NAMPT enzyme activity compared with liver and MBH iNAMPT activity (Line 164–166).

4. Fig.5c and d; Show the data on whether the 2hr-NAMPT treatment is enough to increase cellular NAD⁺ levels. eNAMPT is also known as visfatin, and that can stimulate MAPK and PI3K/AKT pathways, which activations may regulate POMC and/or AGRP gene expressions. This reviewer considers that NMN treatment is enough to investigate this, instead of the NAMPT treatment.

Response: As the reviewer suggested, we examined the effects of NMN treatment on POMC and AGRP transcriptional activity and presented the results in revised Fig. 5c (lower panels). Consistent with NAMPT treatment, NMN stimulated *POMC* promoter activity but suppressed *AgRP* transcriptional activity. Moreover, these effects were inhibited by Sirt2 knockdown.

5. The authors should statistically analyze data by JTK_cycle or related ones, not ANOVA whether daily NAD⁺ or NAMPT levels show circadian patterns. Check this article; Hughes et al., J Biol Rhythms. 2010 Oct;25(5):372-80. doi: 10.1177/0748730410379711.

Response: Reviewer #3 also raised a similar concern on the statistical analysis of circadian rhythms of NAD⁺ and NAMPT levels using ANOVA. Thus, we analyzed the circadian data by using Cosinor analysis and presented the data in Supplementary Tables 1–3.

Minor comments

1. Line 56; Should be “forkhead”.

Response: We corrected the grammatical error in Line 56.

2. Line 84; iNAMPT activity => levels, because the cited article #14 has compared the iNAMPT levels among tissues, not its activities.

Response: We changed to “iNampt expression levels” (Line 87).

3. Which experiments are AtT20 cells used for? There are no descriptions. Also, cell line information is missing in the figure legends.

Response: We apologize for the wrong information on cell lines. In our lab, we usually use AtT20 cells for murine *Pomc* promoter assay. However, in this study, we used SH-SY5Y human neuroblastoma cells in all promoter assays as we measured the promoter activities of human POMC and AGRP genes. N1 cells were used for SIRT2 and FOXO1 interaction study. We added the correct cell line information in Figure legends and revised the corresponding portions in the Methods section (Line 496–497).

4. Line 13; Full-length NAMPT is not a peptide, “protein”.

Response: We changed “NAMPT peptide” to “NAMPT protein” throughout the manuscript.

5. Line 160-161; This reviewer does not understand the sentence, “The negligible effect ...”. Because NAD⁺ amounts in MBH and liver under control conditions are almost the same, 180 pmol/mg tissue, and eNAMPT and iNAMPT function completely in different regions. That suggests i.p. NAMPT should also affect NAD⁺ amount even in the liver. Explain this.

Response: Although the NAD⁺ levels per mg tissue weights are similar in the liver and MBH, the iNAMPT expression levels between the two organs are quite different. In the previous study (Ref. 14), the liver shows a high iNAMPT expression over Actin expression whereas the brain has a low iNAMPT expression. In our assay (Supplementary Fig. 5), the tissue NAMPT activity was higher in the liver than in the MBH. Given higher iNAMPT expression and activity in the liver, additional elevation in liver NAMPT expression by exogenous NAMPT administration may be not significant as shown in Supplementary Fig. 7 and described in Line 171–172.

6. Line 237; Based on the authors’ experiment, they cannot say “direct” binding.

Response: We changed “direct binding” to “molecular interaction” in Line 271–272.

7. Line 238; “acetylated form of FOXO1” is better instead of “expression of acetylated FOXO1”.

Response: We changed to “the expression of acetylated FOXO1” in Line 273.

8. No description of Fig.7b.

Response: We added the description on Fig. 7c (revious Fig. 7b) in Line 322.

9. Fig.1d; Compared to iNAMPT, for this reviewer eNAMPT seems to possess circadian post-translational modifications. This reviewer recommends the authors discuss this.

Response: We discussed the possibility of post-transcriptional modification of eNAMPT in the Discussion section (Line 398–400)

10. Fig.2 b and c; black and white bars are opposite.

Response: The black and white bars are properly allocated but the ZT time was misarranged. We revised the figures and bar graphs. Thank you for the keen comment.

11. Fig.2e and h; iNAMPT amounts in the liver are decreased by siNampt treatment, but why are liver NAD⁺ amounts not changed? Discuss this.

Response: In our previous assay, we measured NAD⁺ amounts in a small piece of liver block. As the liver is quite a big organ, this measurement may be inaccurate. We therefore reassayed liver NAD⁺ levels in pooled liver samples obtained from 4 liver lobes and found that liver NAD⁺ levels were significantly lower in siNampt-injected mice than in siControl-injected mice (revised Fig. 2i).

12. Fig.2 c and g; In controls, Fig.2c shows high EE during the daytime, but Fig.2g shows the opposite result.

Response: The day/night information in Fig 2g was right but that in Fig. 2c was wrong. We corrected this error in revised Fig. 2b, 2c. Thank you for pointing out our mistake.

13. Fig.5c and d; To avoid the possibility that SIRT1 is associated with POMC/AGRP gene expressions, it would be better to knock down Sirt1 and check these gene regulations.

Response: We performed the SIRT1 knockdown study as suggested. As shown in Supplementary Fig. 12, Sirt1 depletion blocked the effects of NAMPT on POMC and AgRP promoter activity. So we revised the description of SIRT1 in the Result (258–263) and Discussion section (448–457).

14. Fig.7a; MBH and plasma NAD⁺ amounts in DIO mice seem to oscillate for this reviewer. Perform statistical analyses.

Response: We analyzed MBH and plasma NAD⁺ levels in DIO mice using cosinor analysis. The circadian rhythm parameters are shown in Supplementary Table 3.

15. No description of per2::luc mice in “Animals”/METHODS.

Response: We described the source of PER2::Luc mice in the “Animals” paragraph of the Method section (Line 491–492).

Thank you again for reviewing our study in detail and providing helpful comments. We hope that our responses and the corresponding revisions are satisfactory.

Reviewer #3 (Remarks to the Author):

This is an interesting paper in which the authors explore the role of eNAMPT in the hypothalamic control of energy balance. Using behavioral, physiological and molecular techniques, the authors found that the eNAMPT-NAD⁺ pathway is very important in the hypothalamic control of energy balance. Although similar results had been shown previously (Yoon et. Al. (2015), in this work the authors propose a specific mechanism in POMC and AGRP neurons, on this control. The authors demonstrate that the NAD⁺-dependent activity of SIRT2 is of central importance in controlling the activity of the transcription factor FOXO1, which in turn would control the expression of POMC and AGRP by inhibiting and inducing their expression respectively. The authors further explored NAD⁺ levels in DIO mice, arguing that in this condition NAD⁺ oscillation lost robustness. Finally, they explored eNAMPT levels in the blood of lean and obese subjects. Although these results are not conclusive, lean subjects appear to have better oscillations in eNAMPT. Although the paper is extensive and used a wide variety of techniques some results seem to be not very conclusive as they are, therefore, I suggest reorganizing the figures and results and adding some analysis including feeding and analytical reinterpretations and also improving the discussion section

Please consider the following observations.

LINE 75 as for definition zt0=lights (at 8am) this info can be included in methods

Response: Thank you for reviewing our study in detail and providing critically helpful comments. We added this information in the Methods section (Line 484–489).

LINE 90 “These findings suggest that a rise in hypothalamic NAD⁺ contents during the dark period may be driven by plasma eNAMPT, whereas its early-morning rise may be driven by MBH iNAMPT.” It’s not clear why the authors suggest this.

Response: As our data did not provide direct evidence, we revised the sentence as follows:

“These findings led us to speculate that a rise in MBH NAD⁺ levels during the dark period may be related to the nocturnal rise in plasma eNAMPT” (Line 94–95).

LINE 97 “In the epididymal adipose tissue, diurnal fluctuations of iNAMPT expression were less distinct than those in the liver and plasma of freely-fed male mice” this is not clear, similar to blood and liver? It’s seem the opposite, a reduction during the dark phase

Response: We reanalyzed circadian fluctuations of adipose iNAMPT expression using cosinor analysis. As shown in Supplemental Table 1, adipose iNAMPT had the opposite circadian oscillation to those in liver iNAMPT and plasma eNAMPT. We revised the sentence accordingly (Line 100–102).

LINE 124 “EE, and FI. eNAMPT inhibition was induced by continuous infusion of FK866” how to know if the iNAMPT was not inhibited as well?

Response: We measured the MBH iNAMPT activity in FK866-infused mice. As shown in Supplementary Fig. 4d, systemic FK866 treatment did not significantly affect the hypothalamic iNAMPT activity (Line 141).

LINE 125 “In mice receiving FK866 infusion (8 mg/kg), nocturnal rises in the MBH NAD⁺ levels” how did the authors calculate this concentration?

Response: For NAD⁺ measurement, we measured the wet weights of MBH blocks, lysed and subjected to HPLC as described in the Methods section. The amount of MBH NAD⁺ was normalized to the tissue weight. The NAD⁺ levels were not concentration and they were pmol (amount) per mg tissue.

LINE 126 “were abolished and the normal circadian patterns” with only two ZT points it is difficult to assume a circadian pattern.

Response: As suggested, we measured MBH and liver NAD⁺ levels every 4 h over the 24-h period. The data are shown in revised Fig. 2d.

LINE 128 “These data suggest that normal diurnal variations in hypothalamic NAD⁺” this is by humoral NAD⁺ levels?

Response: In our NAD assay, we lysed the cells with 150 μ l of 1 M HClO₄. Thus, we assume that hypothalamic NAD⁺ are mostly intracellular NAD. However, strictly speaking, our assay could not differentiate intracellular and extracellular NAD⁺.

LINE 131 “markedly blunted the nighttime increases in LMA and EE”. The reduced effect in EE is by the reduce LMA? Please discuss this.

Response: As LMA is a component of EE, these changes in EE can be explained by the effects on LMA. In addition, eNAMPT could affect EE through the modulation of other components of EE such as thermogenesis considering that POMC and AgRP neurons can also regulate thermogenesis. We added this issue in the Discussion section (Line 416–420).

LINE 132 “during the dark” dark period

Response: We revised it as suggested (Line 137).

LINE 136 “siNampt (33 nmol/kg) in the tail vein reduced plasma eNAMPT by ~40% and MBH NAD⁺ levels” what tissue could be affected in the salvage pathway? WB experiments in the liver, hypothalamus and adipose tissue would be required

Response: We presented the western blot data of tissue iNAMPT in Fig 2j.

LINE 142 “siNampt-induced depletion of blood eNAMPT levels recapitulates the effects of chemical NAMPT inhibition on LMA and EE” What is known about the effects of NAD⁺ on activity, feeding and EE? because feeding is rather opposed to EE and activity (excepting the activity induced prior to feeding in a RF paradigm known as Feeding anticipatory activity)

Response: According to our previous papers (Roh et al., *Metabolism*, 2018, 88: 51–60, Roh et al., *Obesity*, 2018 26, 1448–1456), NAD supplement increased EE/LMA but suppressed food intake through increased hypothalamic NAD⁺ levels .

LINE 147 “dark” dark period

Response: We revised it as suggested (Line 148).

LINE 153 “daytime” rather light phase

Response: We removed “during daytime” in the title (Line 160–161) as it nighttime treatment also increased hypothalamic NAD⁺ levels and locomotor activity.

LINE 155 “endogenous plasma eNAMPT activity” activity? or levels? the authors should provide a reference/figure

Response: We revised the sentence: “*when endogenous plasma eNAMPT levels reach a trough*” (Fig. 1a and 3a) (Line 163–164).

LINE 167 “On the other hand, intraperitoneally injected NAMPT/NMN and ICV-administered NAMPT suppressed fasting-induced hyperphagia” it should be discussed the role of NAD/NMN on energy balance.

Response: We inserted the following sentence in Line 189–191:

“These data indicated that NAMPT/NMN-induced elevation in hypothalamic NAD⁺ levels could lead to a negative energy balance through the stimulation of LMA/EE and suppression of FI.”

LINE 168 “These findings support that circulating eNAMPT is critical for generating the day-night rhythms in LMA and EE.” this is not clear because there are not experiments on the dark phase

Response: In this experiment, we artificially elevated eNAMPT levels during the light period by administering NAMPT protein to the level of endogenous eNAMPT levels at ZT18, as shown in Supplementary Fig. 6, in order to observe changes in LMA and EE.

We also revised the sentences as follow in Line 183–187:

“Artificial elevation in hypothalamic NAD⁺ levels by eNAMPT and NMN administration in the light phase led to increased LMA and EE, which was consistent with the physiological changes in LMA and EE during the dark period. Thus, these findings support the hypothesis that circadian fluctuation in blood eNAMPT and hypothalamic NAD⁺ levels may contribute to the day-night rhythms of LMA and EE.”

LINE 193 “The percentages of c-Fos+ POMC neurons were significantly different between ZT6 and ZT18 (13% vs. 28%, P = 0.04)”. POMC is an anorexigenic peptide. How to explain the increased activity of these neurons at ZT18 vs ZT6

Response: Animals usually consume a large portion of nocturnal food intake in the early hours of dark period (ZT12–16). Thus, the POMC neuronal activity was high at ZT18 (6 h after light-off), which may contribute to satiety formation.

LINE 194 “variation” add “in the activity”

Response: We revised the sentence as suggested (Line 216).

LINE 195 "while the high POMC neuronal activity at ZT18 was suppressed by FK866". How relates this with the natural rhymes in the hypothalamic NAD⁺?

Response: POMC neuronal activity and hypothalamic NAD⁺ levels were high at ZT18 and both factors were decreased by FK866 administration. We therefore added the following sentence in Line 241–243:

“These changes in POMC neuronal activity were consistent with the changes in hypothalamic NAD⁺ levels (i.e. decrease by FK866 and increase by NAMPT) (Fig. 2d and 3b).”

LINE 203 "POMC end product α -melanocyte stimulating hormone (α MSH) and AGRP regulate LMA and EE, and these effects may be partly via their agonistic and antagonistic actions on melanocortin 3 and 4 receptors (MC3/4R)³⁴" more references are needed

Response: We added new references (ref. 35–37) and revised the sentence as follows in Line 226–228:

“As POMC end products α -melanocyte-stimulating hormone (α MSH) and AgRP are each endogenous agonist and antagonist of melanocortin 3 and 4 receptors (MC3/4R)³⁴, POMC/AgRP neurons are thought to regulate LMA and EE via these receptors³⁵⁻³⁷.”

LINE 213 "the circadian oscillations of hypothalamic SIRT1 expression" the circadian expression of Sirt1, has not been clearly demonstrated, rather its activity induced by the oscillating NAD⁺ provokes circadian oscillation in its deacetylase activity and therefore on the acetylation levels on its target's proteins. Therefore, the result should be taken with cation. The authors might discuss this.

Response: The reviewer #2 also suggested to test the role of SIRT1 on eNAMPT-mediated regulation of POMC and AgRP transcriptional activity. As shown in Supplementary Fig. 12, SIRT1 knockdown significantly blocked the effects of NAMPT treatment on POMC and AgRP promoter activity. Thus, we revised the description of SIRT1 in the Result (Line 258–263) and Discussion (448–457).

“

LINE 214 "marked diurnal rhythm in SIRT1 expression that peaked at ZT6–10 and decreased at ZT18"
To get a more precise analysis, a cosinor analysis should be done, to obtain the acrophase and amplitude parameters.

Response: According to the reviewer's suggestion, we conducted cosinor analysis of MBH SIRT1 circadian rhythm and presented the data in Supplementary Table 2.

LINE 219 "Sirt2/NPY-GFP confirmed Sirt2 expression in both types of neurons (Fig. 5b), and in vitro promotor assay revealed that NAMPT treatment increased the POMC transcriptional activity but repressed the AGRP transcriptional activity" since SIRT2 is a deacetylase enzyme, what could be the epigenetic mechanism?

Response: As shown in Fig. 6, SIRT2 regulates POMC and AgRP expression via the deacetylation and inhibition of FOXO1.

LINE 227 "These findings suggested that hypothalamic SIRT2 acts downstream of eNAMPT in the diurnal modulation of LMA and EE involving POMC and NPY/AGRP neurons"

1) what is the effect of NPY or POMC on the circadian activity?

Response: Upon close review, we found that our data do not directly support this notion. We therefore revised the sentence as follows in Line 255–257:

“~hypothalamic SIRT2 acts as a downstream mediator for the role of eNAMPT in the modulation of LMA, EE, FI, and the expression of Pomc, Npy, and Agrp.”

2) the oscillation of POMC or NPY/AGRP should be shown

Response: As suggested by the reviewer, it will be worth showing the circadian oscillation of Pomc and Npy/Agrp transcript levels. To our regret, the number of animals per day, that we could inject siSirt2 into the bilateral ARC and then insert icv cannulae, was limited. So it was almost impossible to have larger numbers of groups in this experiment that was required for examining circadian oscillation. So we could not perform the experiment suggested by the reviewer.

3) food intake is an very important parameter it should be shown,

Response: We added the food intake data in Supplementary Fig. 11.

4) It is not clear the inter or extra cytosolic NAD⁺ mechanism such as the extra to intracellular transportation, etc., need to be discussed.

Response: We discussed the issue of intra- vs extracellular NAD⁺ in the third paragraph of the Discussion section (Line 364–381).

LINE 238 “decreases the expression” rather the acetylation level

Response: We revised the sentence as follows in Line 272:

“Moreover, in N1 hypothalamic neuronal cells, Sirt2 overexpression decreased the expression of acetylated FOXO1.”

LINE 244 "Overexpression of FOXO1 deacetylation mutant (6KR)⁴²"

what is the functional importance of this region/residue for the FOXO1 activity?

Response: In FOXO1-6KR mutant, six lysine residues corresponding to the proposed FOXO1 acetylation sites (K242, K245, K259, K262, K271, and K291) were replaced with arginine to prevent acetylation (ref. 45). The positive charge of these lysines in FOXO1 contributes to its DNA-binding and the acetylation state of these lysine residues has been shown to alter the DNA binding ability of FOXO1

(ref. 47). Moreover, FOXO1 acetylation also affects the phosphorylation of FOXO1 (at Ser 253) by PI3K-Akt signaling, which promotes the nuclear export of FOXO1 and subsequent ubiquitin-mediated degradation. We briefly described this in Line 284–285, 286–287.

LINE 244 “deacetylation mutant” rather with a mutation on the amino acid that is deacetylated by SIRT2 (ref)

Response: We revised the sentence and added the appropriate reference (ref. 43) in Line 290.

LINE 245 “effects of WT FOXO1 on AGRP and POMC transcriptional activity (Fig. 6d).”

This is confused, because it is not possible to differentiate between cellular and recombinant FOXO1 proteins, and how unacetylated FOXO1 decreases the level of FOXO1?, also how to know that the acetylation levels are different? it would be needed an antibody against acetylated residues.

Response: As suggested in a previous study (ref. 47), the FOXO1 acetylation state affects FOXO1 phosphorylation that leads to 14-3-3-mediated nuclear export of FOXO1 and subsequent ubiquitin-mediated degradation. Although we could not differentiate cellular FOXO1 and recombinant FOXO1, we found that FOXO1-6KR overexpression decreased FOXO1 acetylation in cellular total protein extract (Supplementary Fig. 14a) by using immunoprecipitation with FOXO1 antibody followed by immunoblotting with acetylated lysine antibody. In this experiment, we treated cells with MG132 to prevent degradation of deacetylated FOXO1.

LINE 254 “hypothalamic FOXO1 levels (Supplementary Fig. 7), suggesting that eNAMPT can promote hypothalamic FOXO1 degradation in vivo.” to include a reference on these mechanisms mediated by NAMPT/NAD or acetylation

Response: We added the reference and revised the sentence as follows in Line 302:

“...through NAD⁺/SIRT-mediated FOXO1 deacetylation^{45,47}.”

LINE 257 “decrease in hypothalamic FOXO1 levels” at what ZT?

Response: We sacrificed the mice at ZT6 (Line 304).

LINE 271 “blunted diurnal oscillation in MBH NAD⁺ contents, plasma eNAMPT, and liver iNAMPT expression levels (Fig. 7a).” The NAD⁺ levels from control diet feed mice should be included in the same plot also to include a detailed circadian analysis such as the cosinor analysis.

Response: As suggested, we provided the data of lean and obese mice together in Fig. 7a-c with cosinor analysis in Supplementary Table 3.

LINE 278 “The demographic characteristics of the subjects”

These are not demographic characteristics

Response: We removed the word “demographic”.

LINE 280 “Blood samples were collected every 30 min from 7 a.m. to 9 p.m., and then hourly until 7 a.m. in the next morning” There was a period of adaptation on this schedule?

Response: As described in the Methods section on “human eNAMPT study” (Line 699–700), the subjects were advised to sleep between 10 p.m. and 6 a.m. for 1 week before the study.

LINE 287 “showed altered circadian patterns in serum eNAMPT levels with oscillations during the light-off period (Fig. 7d)”

It could be that the effect on the diurnal peaks compared with the night period is rather an effect on the sampling (day vs night).

Response: We plotted the eNAMPT data during the light-on period every 1 h to match the eNAMPT data during light-off periods. As shown in the figure below, we could still see differences in eNAMPT oscillation between the light-on and light-off periods. Therefore, the difference in daytime and nighttime

eNAMPT oscillation may not have been caused by differences in the sampling intervals. Moreover, blood was drawn from an in-dwelling intravenous catheter to avoid stress of blood sampling.

LINE 292 "conditions, which might underlie the disruptions in the diurnal rhythms of physical activity or sleep/wake cycle in obese individuals"

The results as these are, do not support this conclusion additional data analysis should be done such as a cosinor analysis to extract the amplitude, acrophase and mesor parameters, and then to statistically compare between groups.

Response: As you pointed out, in humans, oscillations in plasma eNAMPT levels during light-on period looked like ultradian rhythms. Notably, there was a near-complete absence of oscillation during the light-off period and this day-night difference may represent a circadian pattern of fluctuation in plasma eNAMPT concentrations as we discussed in Line 380–384. Therefore, cosinor analysis seemed to be inappropriate to analyze this pattern of oscillations in human serum eNAMPT and to compare between lean and obese human subjects.

We revised the sentence as follows in Line 339–342:

“Although our data were obtained from a small number of young healthy subjects, these human data suggest that day-night difference in blood eNAMPT oscillations can be altered under chronic overnutrition conditions and this phenomenon might relate to altered day-night rhythms of physical activity in obese individuals^{50,51}”

LINE 295 “This rhythm persists under continuous darkness and is profoundly disturbed by the reversal of the light-dark phase and daytime-restricted feeding”

This is weird, because the authors restricted the feeding during the day being in fact the sleeping timing in mice, therefore I would rather expect a blunted effect when the feeding is restricted during the dark period, therefore I suggest restricting the food when the mice are active.

Response: Considering that rodents consume a large portion of food intake during the dark period, in our food shift study, we restricted food consumption during the light period in order to investigate the effect of food shift to daytime on the circadian rhythms of plasma eNAMPT–hypothalamic NAD⁺.

LINE 299 "Supporting this possibility, eNAMPT has a high enzymatic NAMPT activity¹⁴ 299 and its blood levels rise during the night"

Please discuss what may be the source of the hypothalamic eNAMPT.

Response: We discussed the source of the nocturnal rise in eNAMPT in the Discussion section (Line 382–405).

LINE 302 "These findings strongly suggest that eNAMPT-mediated regulation of hypothalamic NAD⁺ biosynthesis occurs in a circadian manner"

To be a strong suggestion the data should be taken at more ZT points, so I recommend eliminating "strong"

Response: We removed "strongly" from the sentence (Line 351).

LINE 304 "diurnal fluctuation in 6-months-old mice although the degree of fluctuation was less than that in our study." Do the authors mean amplitude for the degree of fluctuation?

Response: We changed to "amplitude" in Line 355.

LINE 305 "old mice" when compared...

Response: We revised the sentence accordingly (Line 355).

LINE 308 "Consistent with the findings from rodents, humans also display a distinct diurnal rhythm in blood eNAMPT"

What means for distinct, compared to what?

Response: We removed the word "distinct" from the sentence (Line 357–359).

LINE 310 "Frequent blood sampling in our human study involving young lean subjects also revealed pulsatile oscillations during the day"

These could be ultradian rhythms and not circadian rhythms, this should be discussed and/or reanalyzed

Response: As you commented, eNAMPT oscillations during the light-on period seem to show ultradian rhythms with 3 to 7 peaks. We discussed this in the Discussion (359–363).

LINE 311 this looks like a result description rather than a discussion

Response: We revised this sentence as follows in Line 361–363:

"Notably, there was a near-complete absence of oscillation during the light-off period and this day-night difference may represent a circadian pattern of fluctuation in blood eNAMPT concentrations".

LINE 303 "There is compelling evidence that the eNAMPT upregulates hypothalamic NAD⁺ 312 levels, but the detailed mechanisms remain elusive" Please include references and the participation of eNAMPT on the NAD biosynthesis should be discussed in more detail.

Response: We added the appropriate references (#13, #15) and discussed how eNAMPT could affect hypothalamic NAD⁺ biosynthesis in the third paragraph of Discussion (Line 364–381).

LINE 339 "Future studies are needed to examine whether hepatocytes secrete eNAMPT through sirtuin- and EV-dependent mechanisms as shown in adipocytes^{13,15}"

This data seems to contradict the contribution of adipose tissue on the hypothalamic levels of NAMPT shown by Yoon et. Al. (2015). How to explain this discrepancy?

Response: We discussed the possible differential roles of adipocyte- and hepatocyte-derived eNAMPT in the regulation of physical activity under different physiological conditions (i.e., fasting condition and nocturnal freely-fed condition) (Line 406–415).

LINE 348 "Furthermore, IP administration of NAMPT peptide and NMN, which imitated the condition

of increased systemic eNAMPT activity “

At which time was injected the NMN and NAMPT peptide? does this peptide has enzymatic activity?

Response: We injected NAMPT protein and NMN at ZT5 as shown in Fig. 3a. The enzymatic activity of NAMPT protein is provided in the company website (<https://adipogen.com/ag-40a-0018-nampt-visfatin-pbef-human-rec-his.html>) and also tested in our hand (Supplementary Fig. 5).

LINE 350 “In contrast, eNAMPT appears to have a dispensable role in regulating the physiological fasting/feeding rhythms”. Also, here this data looks contradictory with the data reported by Yoon et. Al. showing that eNAMPT its important to maintain hypothalamic NAD⁺ levels during fasting

Response: We discussed the discrepancy in Line 406–412.

LINE 352 “Instead, eNAMPT may be involved in satiety formation in the middle of the dark phase, considering that systemic eNAMPT inhibition with FK866 increased food intake during the night“

What can be discussed about gender? Also, as already suggested a food intake experiment would be very relevant; such as circadian or at least 2-4 ZT food consumption and/or the measurement of food intake during fasting at night and at day periods

Response: We discussed the gender issues (Line 412–415) and added the corresponding sentence regarding food intakes (Line 422–424).

LINE 376 “These findings support the hypothesis that eNAMPT stimulates LMA and EE through SIRT2-mediated FOXO1 inhibition and the resulting functional changes in POMC and NPY/AGRP neurons”. It could be very nice if the authors discuss in more detail the mechanism of FOXO1-mediated regulation of POMC and NPY neurons

Response: We and others (ref. 41, 42) have previously shown that FOXO1 binds the AgRP and NPY promoters to simulate the transcription of AgRP and NPY. On the other hand, FOXO1 inhibited STAT3-stimulated POMC transcription. We described this issue in Line 437-440.

LINE 381 “In addition, we found that DIO mice had blunted circadian oscillations in the liver iNAMPT and plasma eNAMPT expression and MBH NAD⁺ levels”

This is not clear, I suggest to directly compare these oscillation under control and HFD and/or analyze the circadian parameters

Response: As the reviewer suggested, we compared the circadian parameters of the liver iNAMPT, plasma eNAMPT expression, and MBH NAD⁺ levels between CD-fed mice and HFD-fed mice and showed the results in Fig. 7b and Supplementary Table 3.

LINE 382 “Moreover, DIO disrupted diurnal rhythms in hypothalamic FOXO1 expression and LMA”
As in the last suggestion, to analyze the circadian parameters on the levels of FOXO1

Response: We also analyzed the circadian parameters of hypothalamic FOXO1 in CD-fed mice and HFD-fed mice. The data are shown in Supplementary Table 3.

FIGURE 1B To separate light/dark areas as done in the panel A

Response: This experiment was conducted under continuous darkness. We added “continuous darkness” in the schematic diagram of Fig. 1b to avoid confusion.

FIGURE 2A In liver NAD⁺ are increased at the night onset, how to explain the increase in the NAD⁺ levels when eNAMPT is inhibited?

Response: As reviewer #2 also raised concern on this data, we reanalyzed the circadian oscillations of hepatic NAD⁺ levels during FK866 treatment by using pooled liver samples and by collecting liver samples at multiple time points. As shown in revised Fig. 2d, FK866 treatment significantly decreased the liver NAD⁺ levels.

FIGURE 2H Are there any WB for these results?

Response: We presented the WB images along with graphs in Fig. 2j.

FIGURE 3A What about the dark period?

Response: To address the reviewer's question, we injected NAMPT protein and NMN during the dark period (ZT17) and measured the MBH NAD⁺ contents at ZT18. The data are now shown in supplementary Fig. 8 and the results are described in Line 171–177.

FIGURE 4A Mesor is not the amplitude. These plots are not clear, blue=vehicle and red=NAMPT, but in the histograms, they are grouped in pairs

Response: We corrected the errors in revised Fig. 4a.

FIGURE 6B What means -OX?

Response: Ox refers to overexpression. We removed Ox and presented the experimental scheme in revised Fig. 6b.

FIGURE 6B SIRT1 inhibits FoxO1-mediated repression/expression of POMC / AGRP, respectively. does this match with the NAD⁺ levels maybe to show a plot

Response: We assume that this comment is related to Fig. 6C and SIRT2 (not SIRT1). As SIRT2 and FOXO1 are downstream of the NAMPT-NAD⁺ axis, overexpression of FOXO1 and SIRT2 is expected not to alter the cellular NAD⁺ levels.

FIGURE 6B To include the molecular weights

Response: We included the molecular weights on all WB blots.

FIGURE 6D In this experiment SIRT1 over-expression would be included

Response: Fig. 6d shows the effects of overexpression of FOXO1-WT and WT-6KR on POMC and AGRP transcription. We assume that this suggestion may be related to Fig. 6c. We thus investigated the effect of SIRT1 alone overexpression and presented the data in Supplementary Fig. 13. Like SIRT2, SIRT1 overexpression antagonized the effects of FOXO1 on POMC and AGRP transcriptional activity.

Thank you again very much for reviewing our study in detail and providing helpful comments. We hope that our responses and the corresponding revisions are satisfactory.

Reviewers' Comments:

Reviewer #1:

Remarks to the Author:

This is a revised manuscript. In the current manuscript, the authors have provided new data, corrected the errors and inaccurate writings. Overall, the questions that I raised for the original manuscript have been satisfactorily addressed. I have no further concerns about this manuscript.

Reviewer #2:

Remarks to the Author:

1. The authors used "Cosinor analysis" to check whether fluctuations are circadian oscillations or not, instead of "JTK cycle." This reviewer thinks that Cosinor analysis is used to analyze periods, amplitudes, etc. of samples that show circadian oscillation, not to confirm whether samples show oscillations. For example, this reviewer agrees that SIRT2 does not seem to oscillate (Fig. 5a), but how does Cosinor analysis in Table S2 support that?

2. "the expression of acetylated FOXO1" in line 273; "the acetylated form of FOXO1" is better than that.

Reviewer #3:

Remarks to the Author:

The manuscript has been significantly improved, I consider it worth publishing in Nature Communications, in consideration of the following suggestions

Line 80 "and a trough during ZT6-10 (mid-to-late light phase)" this is not clear, is that period a peak?

Line 116 "without food restriction (light-shift)" Rather ad-labium

Also "light-shift and food-shift is confusing rather something like light-dark entrainment of food-entrainment or light-dark zeitgeber or food as zeitgebers

Line 117 to include ZTs (ZT0-ZT12)

Line 118 "The

"circadian variations in " rather circadian fluctuation/oscillation

"group but adjusted to shifted light-dark cycle (Fig. 1i)" this is not clear because both light at subjective day or subjective night show similar increased levels of blood eNAMPT, therefore eNAMPT seems not to be altered by "shifted light dark cycle", "adjustment to lighth-dark cycle" should imply a reversal of eNAMPT levels.

Figure 1i Where is the WB for this quantification?

Line 119 Here is some confusing to suddenly change from blood ENAPT to MBH NAD+ the authors should add more context

Line 121 Perhaps mention the MBH result first and then the liver, as Figure 1 ends with MBH NAD+ levels.

Line 123 Rather light and food as zeitgebers in iNAPT are important in MBH while in peripheral tissues it is food.

Line 131 The reference from Myeong Jin Yoon should be included here

Line 135 LMA is only show at two time points, to understand if LMA is related with EE or food intake, more time resolution is requiered (e.g) hourly plots in LMA and EE

Line 136 According to the authors, inactivation of NAMPT reduces AML and EE and induces food intake. Since food intake requires LMA, further analysis would be necessary, e.g., measurement of food anticipatory activity (FEA) would help to specify whether the reduction in LMA is indeed related to a reduction in EE.

Line 138 Why it is not mentioned the figure 3B?

Line 140 Since NAMPT is directly cotrolled at the transcriptional level by the molecular clock, mRNA expression levels should be shown, in order to analyze whether NAMPT activity is due to its transcriptional level or to posttranslational mechanisms.

Line 144 "Circadian variations in hypothalamic NAD+ levels" this might be circadian biosynthesis

Line 163 injected recombinant NAMPT protein

Line 192 What could be the hypothalamic mechanism that reduces food intake by NAD+?

Line 203 There are reports that sirt1 in the SCN is important for clock control (Cell. 2013 Jun 20;153(7):1448-60), please discuss this discrepancy.

Line 214 What is the relationship of this patten expression to that of feeding?

Line 238 In Figure 5A in the graph both notifications are marked as SIRT2.

Line 283 It is clear from Figure 6C that FOXO1 inhibits and increases POMC and AGRP expression, respectively, but although SIRT2 appears to reduce the effect in the case of POMC expression, this does not appear to be the case for AGRP expression.

Figure 6C right appears that FOXO1 strongly induces AGRP expression even in the presence of SIRT2, the same in the putative Figure 13, FOXO1 reduces and increases POMC and AGRP expression respectively in the precence of SIRT1. Please include a table with statistical analysis or reanalyze the data.

LINE 342 In these results I recommend adding an additional graph such as the area under the curve measuring daytime versus nightttime blood levels,

Line 477 Human recombinant

Line 537 The authors should mention what this system measures (VO2, O2, ETC) and the method (indirect calorimetry).

Responses to Reviewers' comments:

We sincerely thank the reviewers for their helpful comments. We believe that our manuscript has been improved by their advice.

Reviewer #1 (Remarks to the Author):

This is a revised manuscript. In the current manuscript, the authors have provided new data, corrected the errors and inaccurate writings. Overall, the questions that I raised for the original manuscript have been satisfactorily addressed. I have no further concerns about this manuscript.

Thank you again for reviewing our study and providing helpful comments.

Reviewer #2 (Remarks to the Author):

1. The authors used “Cosinor analysis” to check whether fluctuations are circadian oscillations or not, instead of “JTK cycle.” This reviewer thinks that Cosinor analysis is used to analyze periods, amplitudes, etc. of samples that show circadian oscillation, not to confirm whether samples show oscillations. For example, this reviewer agrees that SIRT2 does not seem to oscillate (Fig. 5a), but how does Cosinor analysis in Table S2 support that?

Response: As the reviewer suggested, we conducted a “JTK cycle” analysis for SIRT2. We presented *P*-values of JTK cycle analysis in the Supplementary Table 1, 2, 3.

2. “the expression of acetylated FOXO1” in line 273; “the acetylated form of FOXO1” is better than that.

Response: We revised the sentence on Line 276 as suggested.

Thank you again for reviewing our study in detail and providing helpful comments. We hope that our responses and the corresponding revisions are satisfactory.

Reviewer #3 (Remarks to the Author):

The manuscript has been significantly improved, I consider it worth publishing in Nature Communications, in consideration of the following suggestions.

Line 80 "and a trough during ZT6-10 (mid-to-late light phase)" this is not clear, is that period a peak?

Response: We used trough to mean the lowest level. We changed the sentence on Line 79 for clarity.

Line 116 "without food restriction (light-shift)" Rather ad-libitum

Also "light-shift and food-shift is confusing rather something like light-dark entrainment or food-entrainment or light-dark zeitgeber or food as zeitgebers

Response: We changed “without food restriction” to “ad libitum-feeding” (Line 115). We also added

“light-dark entrainment” and “food entrainment” to the sentences (Line 115, 116) and Fig. 1h.

Line 117 to include ZTs (ZT0-ZT12).

Response: We added “ZT0-ZT12” to Line 116.

Line 118 "The “circadian variations in " rather circadian fluctuation/oscillation "group but adjusted to shifted light-dark cycle (Fig. 1i)" this is not clear because both light at subjective day or subjective night show similar increased levels of blood eNAMPT, therefore eNAMPT seems not to be altered by “shifted light dark cycle”, "adjustment to lighth-dark cycle" should imply a reversal of eNAMPT levels.

Response: We revised the sentence as suggested (Line 117). As the reviewer commented, the rhythm of blood eNAMPT was reversed by the light shift in the circadian time (2 am and 2 pm) data as shown below (Fig. 1i and 1j in the first-submitted paper)

Reviewer #1 suggested that “The authors should use only the ZT for labeling all the figures” to avoid confusion; therefore, we changed the graphs to use only ZT. To address reviewer #1, we show the data in ZT.

Figure 1i Where is the WB for this quantification?

Response: Fig 1i, we show the eNAMPT ELISA data. The concentrations of blood eNAMPT are in ng/ml.

Line 119 Here is some confusing to suddenly change from blood eNAMPT to MBH NAD⁺ the authors should add more context

Response: We added the sentence, “We also investigated whether altered food availability and light-dark cycle also affected the circadian oscillations in hypothalamic NAD⁺/iNAMPT levels.” (Line 118–119).

Line 121 Perhaps mention the MBH result first and then the liver, as Figure 1 ends with MBH NAD⁺ levels.

Response: We changed the order of descriptions on Line 121–123 and Supple Fig. 2.

Line 123 Rather light and food as zeitgebers in iNAMPT are important in MBH while in peripheral tissues it is food.

Response: This sentence included the findings of Fig. 1i, j, and Supple Fig. 2. As the circadian rhythms of blood eNAMPT, hypothalamic NAD⁺, iNAMPT, and liver iNAMPT were blunted or reversed by the food or light shift manipulations. Thus, we revised the sentence as follows: “light and food may act as important zeitgebers in the cyclic fluctuations of hypothalamic NAD⁺, plasma eNAMPT, and MBH/liver iNAMPT levels” (Line 124–125).

Line 131 The reference from Myeong Jin Yoon should be included here

Response: Thank you for your appropriate suggestion. We inserted the reference (ref. 15) (Line 132).

Line 135 LMA is only show at two time points, to understand if LMA is related with EE or food intake, more time resolution is required (e.g) hourly plots in LMA and EE.

Response: We show the hourly plots of LMA and EE in the left panel of Fig 2b, c. The bar graphs show the average values of daytime and nighttime.

Line 136 According to the authors, inactivation of NAMPT reduces LMA and EE and induces food intake. Since food intake requires LMA, further analysis would be necessary, e.g., measurement of food anticipatory activity (FEA) would help to specify whether the reduction in LMA is indeed related to a reduction in EE.

Response: Thank you for your suggestion. We apologize for not showing the FEA data; however, our laboratory does not have the facility for the FEA measurement.

Line 138 Why it is not mentioned the figure 3B?

Response: In the previous manuscript, we mentioned Supplementary Fig. 3B on Line 148. To avoid confusion, we changed Supplementary Fig. 3b to Supplementary Fig. 5 and revised the mention on Line 148.

Line 140 Since NAMPT is directly controlled at the transcriptional level by the molecular clock, mRNA expression levels should be shown, in order to analyze whether NAMPT activity is due to its transcriptional level or to posttranslational mechanisms.

Response: Thank you very much for the thoughtful comment. To address the reviewer’s question, we measured the Nampt mRNA expression in the MBH and liver throughout the circadian cycle. As shown in the following figures, the FK866 treatment did not alter the Nampt mRNA expression in the MBH and liver. Consistently, FK866 has been shown to suppress NAMPT activity via the interaction with the pyridyl group of NAMPT (J. Am. Chem. Soc. 2013, 135, 3485–3493).

Line 144 "Circadian variations in hypothalamic NAD⁺ levels" this might be circadian biosynthesis

Response: We revised as suggested (Line 144).

Line 163 injected recombinant NAMPT protein

Response: We revised as suggested (Line 163).

Line 192 What could be the hypothalamic mechanism that reduces food intake by NAD⁺?

Response: The proposed mechanism of NAD⁺-mediated feeding regulation is that elevated NAD⁺ levels in the hypothalamic neurons increase the SIRT1/2 activity. Activated SIRT1/2 stimulates the transcription of POMC, a precursor of anorexigenic peptide α -melanocyte-stimulating hormone, whereas it suppresses the transcription of orexigenic AgRP and NPY via deacetylation and inhibition of FOXO1 (Fig. 5c, 6c, Supplementary Figs. 14 and 15). As a result, increased anorexigenic Pomc expression and reduced Npy/AgRP expression (Fig. 4c) would lead to a reduction in food intake.

Line 203 There are reports that sirt1 in the SCN is important for clock control (Cell. 2013 Jun 20;153(7):1448-60), please discuss this discrepancy.

Response: As you commented, NAD⁺-iNampt-SIRT1 is known to be a critical component of the SCN clock activity. When we cultured the SCN slices in the FK866-containing medium, the SCN clock activity was markedly reduced (Supplementary Fig. 11). These findings suggested that iNAMPT activity in SCN slices is critical for the clock activity. We described it on Lines 203–205.

Line 214 What is the relationship of this pattern expression to that of feeding?

Response: POMC neurons are activated in the postprandial period to suppress food intake, whereas they are inactivated in a fasted condition. Consistently, the c-fos activity of POMC neurons was lower at ZT6 (mid-day), which is the sleeping period in mice. In this period, mice do not consume much food and would be in a fasted condition. In contrast, the POMC neuron's c-fos activity was increased at ZT18 (midnight), which is the postprandial period. As mice consume a large amount of food in the early dark period (ZT12–ZT18), mice would be in a satiated condition, and POMC neurons were activated at ZT18. We added "according to the circadian fasting (ZT6)–feeding (ZT18) cycle" to the sentence on Lines 218–219.

Line 238 In Figure 5A in the graph both notifications are marked as SIRT2.

Response: We appreciate the reviewer for pointing out our mistake. We corrected our mistake in Fig. 5a.

Line 283 It is clear from Figure 6C that FOXO1 inhibits and increases POMC and AGRP expression, respectively, but although SIRT2 appears to reduce the effect in the case of POMC expression, this does not appear to be the case for AGRP expression. Figure 6C right appears that FOXO1 strongly induces AGRP expression even in the presence of SIRT2, the same in the putative Figure 13, FOXO1 reduces and increases POMC and AGRP expression respectively in the presence of SIRT1. Please include a table with statistical analysis or reanalyze the data.

Response: We apologize for the wrong *P* value information in Fig. 6c and previous Supplementary Fig. 13 (now Supplementary Fig. 15). We reanalyzed the data, included the corrected *P* values, and provided the statistic tables in the letter. Although the statistical power between FOXO1 alone and FOXO1+SIRT2 or between FOXO1 alone and FOXO1+SIRT1 differed in POMC and AgRP transcriptional activities, statistical significance was still found between the groups.

	Comparison groups	P value
POMC transcriptional activity	Mock vs. FOXO1	0.004
	Mock vs. SIRT2	0.012
	FOXO1 vs. FOXO1+SIRT2	0.004
	SIRT2 vs. FOXO1+SIRT2	0.097
	Mock vs. SIRT1	0.002
	FOXO1 vs. FOXO1+SIRT1	0.027
	SIRT1 vs. FOXO1+SIRT1	0.004
AgRP transcriptional activity	Mock vs. FOXO1	0.003
	Mock vs. SIRT2	0.009
	FOXO1 vs. FOXO1+SIRT2	0.047
	SIRT1 vs. FOXO1+SIRT1	0.001
	Mock vs. SIRT1	0.003
	FOXO1 vs. FOXO1+SIRT1	0.004
	SIRT1 vs. FOXO1+SIRT1	0.001

LINE 342 In these results I recommend adding an additional graph such as the area under the curve measuring daytime versus nighttime blood levels.

Response: As you suggested, we calculated the AUC of serum eNAMPT. The obese subjects with BMI > 30 had a higher AUC value of serum eNAMPT compared with that of the lean group in the dark period. In addition, the AUC of serum eNAMPT tended to be higher in these obese subjects than in the lean subjects (Fig. 7f). We described this finding on Lines 342–344.

Line 477 Human recombinant

Response: We changed to recombinant human NAMPT (Line 483).

Line 537 The authors should mention what this system measures (VO₂, O₂, ETC) and the method (indirect calorimetry).

Response: We presented the additional description in the Methods section (Lines 544–546).

Thank you again for reviewing our study in detail and providing helpful comments. We hope that our responses and the corresponding revisions are satisfactory.